

# Alignment of Scanning Lidars in Offshore Campaigns - an Extension of the Sea Surface Levelling Method

Kira Gramitzky[1,2], Florian Jäger[1,3], Doron Callies[1,3], Tabea Hildebrand[1,3], Julie K. Lundquist[4], and Lukas Pauscher[1,3,5]

[1]Fraunhofer Institute for Energy Economics and Energy System Technology (IEE), Joseph-Beuys-Str. 8, 34117 Kassel, Germany
[2]University of Kassel, Department of Integrated Energy Systems, Wilhelmshöher Allee 71-73, 34121 Kassel, Germany
[3]University of Kassel, Department of Sustainable Electrical Energy Systems, Wilhelmshöher Allee 71-73, 34121 Kassel, Germany
[4]Johns Hopkins University, 3400 N. Charles Stree, Baltimore, MD, USA
[5]Vrije Universiteit Brussel, Acoustics and Vibrations Research Group, Pleinlaan 2, Brussels, 1050, Belgium

**Correspondence:** Kira Gramitzky (kira.gramitzky@iee.fraunhofer.de)

**Abstract.** The expansion of offshore wind energy creates an increasing potential for the use of scanning lidars for wind resource assessment, power curve verification, and wake monitoring. These applications require accurate measurements, which require precise alignment calibration. However, performing such calibration offshore is challenging due to the absence of fixed hard targets. The Sea Surface Levelling (SSL) method, which uses the sea surface as a reference, has emerged as a promising

alternative but so far could not determine the static elevation offset and lacked a rigorous uncertainty evaluation. This study presents a generalisation of the SSL method that enables simultaneous determination of pitch, roll, and elevation offset by incorporating scans at multiple elevation and azimuth angles, either with RHI or PPI scans. An analysis of uncertainties is performed, taking into account incorrect distance measurements of the lidar's point of entry into the water, the influence of waves and statistical noise. The extended SSL method is applied to data from a scanning lidar (Vaisala WindCube WLS400S)

installed on the transition piece of an offshore wind turbine in the German North Sea. Results show that pitch and roll can be determined with high confidence and reproducibility, even with incomplete azimuth coverage. For the first time, we demonstrate that elevation offset can be derived directly from SSL, though its accuracy depends strongly on distance determination. Further, correcting the lidar pulse length improves the agreement with an independent validation. Wave effects were negligible under calm conditions but are expected to increase in rougher seas. Overall, the extended SSL method enables the alignment

calibration with typical uncertainties of $0.03°$-$0.04°$ at suitable elevation angles, in this example from $-1.5°$ to $-0.3°$ at a lidar height of around $20\,\mathrm{m}$. The extended SSL method provides a robust, transferable alternative to hard-target calibration for scanning lidars offshore.

## 1 Introduction

The world's offshore wind capacity is growing steadily and forecasts indicate further growth in the coming years (e.g. Global

Wind Energy Council, 2025; Williams et al., 2025). This growth makes offshore wind an important part of the energy transition.





To exploit this potential, it is important to optimise design, layout and operation of turbines and wind farms (Gebraad et al., 2016). An essential basis for this optimisation is on-site wind measurement, which provides important information for resource assessments, site suitability analysis, power performance verification and operational optimisation.

Scanning lidars offer great potential to quantify wind conditions onshore (Käsler et al., 2010; Bodini et al., 2017; Wildmann et al., 2018; Menke et al., 2018) and offshore (Goit et al., 2020; Hildebrand et al., 2025). In contrast to profiling lidars, which measure along a fixed vertical profile, scanning lidars feature a movable scanning head that can direct the laser beam in different directions. This flexibility allows measurements at multiple discrete points or across an area (Vasiljević, 2014; Vasiljević et al., 2016). Depending on the measurement objective, scanning lidars can perform different scanning patterns, for example such as Plan Position Indicator (PPI) scans, which sweep the beam horizontally at a constant elevation angle, Range Height Indicator (RHI) scans, which sweep vertically at a fixed azimuth angle or so-called single Line-of-Sight (LOS) scans. In homogeneous flows, mean wind speeds can also be determined accurately with a single scanning lidar using specific scanning patterns (e.g. Krishnamurthy et al., 2017; Shimada et al., 2020). Combining two or more scanning lidars (dual-Doppler or multi-lidar systems) improved turbulence measurements and spatial coverage compared to profiling lidars (Newsom et al., 2008; Pauscher et al., 2016; Peña and Mann, 2019).

The introduction of scanning lidars into industry-driven wind energy applications is relatively recent. Consequently, several aspects of their calibration, standardisation, and application still require research. Newer long-range scanning lidars are, according to the manufacturer's specifications, designed for distances of over 10 km (e.g. Leosphere (Vaisala), 2015; Abacus Laser GmbH, 2023); however, weather conditions such as fog, low clouds, or very clear air can significantly reduce measurement range and data availability (Rösner et al., 2020). Establishing robust and standardised procedures for alignment, data processing, and uncertainty quantification remains a key area of ongoing investigation.

Installing measurement devices offshore is particularly challenging, and the costs of installing offshore masts are very high. Scanning lidar systems offer key advantages thanks to their long range: they can be operated from coastal locations (Cheynet et al., 2021; Shimada et al., 2022, 2025) or installed directly on existing platforms, like the transition piece of a wind turbine, (Goit et al., 2020; Cañadillas et al., 2022) to perform remote measurements over the water. Furthermore, they enable measurement of wind speed and direction at multiple points or over an entire area. Scanning lidars have already been deployed in various offshore projects, providing data for analysing wake effects (Krishnamurthy et al., 2017; Schneemann et al., 2020; Cañadillas et al., 2022; Anantharaman et al., 2022), quantifying blockage effects within wind farms (Schneemann et al., 2021), performing power curve verification (Gómez et al., 2023), and forecasting power performance (Theuer et al., 2022).

When measuring at distances beyond 1 km, precise alignment of the laser is essential for accurate wind measurements, as even very small angular misalignments can lead to deviations of several meters from the desired measurement point. For example, at a distance of 5 km, a misalignment of 0.1° already translates into a deviation by approximately 8.7 m of the location of the measurement. At a reference height of 100 m with a wind speed of 10 m/s and a typical offshore shear exponent of 0.12, a vertical misalignment of 10 m results in a wind speed bias of approximately 1.15 %, which translates into an error in power estimation of about 3.48 %. To reduce measurement uncertainty, accurate alignment calibration is therefore imperative.



A target accuracy of about $0.02°$ has been shown to be achievable and should therefore be considered the benchmark for alignment calibration (Rott et al., 2022).

Onshore, hard targets are often used for levelling, using the carrier-to-noise-ratio (CNR) mapping method described in Vasiljević et al. (2016). Offshore, it is usually more challenging to find suitable hard targets. However, drones have recently been proposed and tested as mobile hard targets for offshore scan head alignment (Oldroyd et al., 2024). While drone-based

hard target techniques show promise, practical challenges remain, especially related to drone positional uncertainty which can be significant at longer ranges. In recent tests, Oldroyd et al. (2024) reported uncertainties of up to $0.17°$ under high wind conditions, though values closer to $0.05°$ may be achievable in calmer conditions, suggesting that drone-based methods are not yet at the $0.02°$ benchmark but represent a viable alternative where fixed targets are unavailable.

However, an alternative approach to alignment calibration in offshore environment is the sea surface levelling (SSL) method

presented by Rott et al. (2022, 2021a, b), which uses the sea surface as a reference instead of hard targets. SSL is based on the idea, that when ideally aligned, the distance between the lidar and the sea surface remains constant at a constant elevation angle in all directions (azimuth angles), assuming a flat sea surface. Deviations in the distance can be used to determine and correct the inclination of the device. Rott et al. (2022) performed SSL using PPI scans. Measurements were conducted with a long-range Doppler scanning lidar (WindCube 200S, Leosphere/Vaisala) installed on the transition piece of a wind turbine in

the German North Sea for several months from 2018 to 2019. Comparison with a model describing the turbine's inclination under wind thrust showed promising results, with the lidar alignment achieving a very high accuracy and a levelling uncertainty of approximately $0.02°$. Champneys et al. (2024) and Gramitzky et al. (2024) present an alternative SSL approach based on RHI scans at different distances. In comparison with co-located inclinometers, Gramitzky et al. (2024) showed that both the PPI and RHI methods perform well, with only minor differences in estimating pitch and roll angles.

Two important knowledge gaps in SSL prevent its widespread use. First, the SSL methods described in Rott et al. (2022) and Gramitzky et al. (2024) cannot determine the static elevation offset - a constant deviation of the laser beam in all directions - which is required to determine the exact alignment of the laser beam in addition to pitch and roll. Second, although existing studies demonstrate the potential for highly accurate pitch and roll estimation, no rigorous method for uncertainty quantification in SSL has been presented so far. This knowledge gap is especially important for resource assessment and power curve

verification, where rigorous uncertainty quantification and reduction are paramount.

The aim of this work is to fill these gaps by:

1. generalizing the SSL method to determine the elevation offset, in addition to pitch and roll, and

2. establishing a rigorous method to quantify the uncertainties for the alignment calibration derived from SSL.

For this purpose, this paper presents the extended SSL method that incorporates scans at multiple elevation and azimuth

angles into the SSL process, rather than relying on a constant distance or fixed elevation angle. In a first step a rigorous method for estimating uncertainties in pitch, roll, and elevation offsets obtained from SSL is developed, explicitly accounting for uncertainties in the distance to the sea surface, wave-induced variations, and statistical uncertainties from SSL fitting. To investigate the influence of waves, a simple wave model is used to examine the influence of wave-induced fluctuations





on measurement uncertainties. The sensitivity of these uncertainties to the selection of scan angle ranges is analysed, and
recommendations for SSL configurations that minimize overall measurement uncertainties are provided. Then, this novel SSL
approach is experimentally evaluated for different setting using measurement data obtained from a WindCube WLS400S
(Vaisala) installed on the transition piece of a wind turbine in an offshore wind farm in the North Sea. Validation is conducted
by comparing the SSL results with pitch and roll measurements from an external inclinometer and the elevation offset with
hard target calibration performed by an external consultancy mapping results obtained from a fixed hard target.

The structure of the paper is as follows. Section 2.1 introduces the variables and equations that describe the alignment
of scanning lidars, while Sect. 2.2 presents the SSL theory and its generalisation. Section 2.3 explains how the distance to
the water surface is determined using the CNR-over-range signal, and Sect. 2.4 introduces the wave model. Sections 2.5 and
2.6 cover the measurement campaign and data preparation. Section 3.1 examines the results of the modelled measurement
uncertainties, and Sect. 3.2 shows the analysis of SSL results from the scanning lidar measurement campaign in the North Sea
for different settings. Finally, Sect. 4 provides a discussion, and Sect. 5 concludes the paper with a summary and key findings.

## 2 Methodology

### 2.1 Alignment of scanning lidar

The general purpose of an alignment calibration of a scanning lidar is to derive a function that allows the transformation
of coordinates from the internal coordinate system of the lidar to an external external orthogonal coordinate system. The
transformation matrix must be known in order to accurately target the correct measurement point and to determine the location
for each measurement. The desired measurement points are usually defined in an external orthogonal coordinate system aligned
with the vertical, south-north and west-east axes. The difference between the coordinate systems arises from multiple sources.
Firstly, the lidar device is often not perfectly aligned with true north and may not be levelled during its setup, resulting in
a tilt. Secondly, the scanning mechanism may exhibit deviations in the vertical lidar axis. This so-called elevation offset is
independent of the azimuth angle and causes the laser beam to exit consistently higher or lower in all directions (e.g. Vasiljević
et al., 2016). Possible causes include misalignment of the mirrors within the scan head. These factors are important to ensure
accurate measurements and reliable data from scanning lidar measurements.

The measurement location of a scanning lidar in the internal coordinate system is usually defined by the elevation angle
$\varphi_\mathrm{L}$, the azimuth angle $\theta_\mathrm{L}$ and the measurement range from the lidar $r$. Herein, the azimuth angle is defined clockwise from
the y-axis with device north corresponding to $\theta_\mathrm{L} = 0°$. The elevation angle $\varphi_\mathrm{L}$ is defined relative to the internal horizontal
plane, which is spanned by the device's internal north and east vectors. Negative angles mean that the laser beam is pointing
downwards (see Fig. 1a). The range $r$ from the lidar is defined by the time of flight of the laser pulse. The returned signal
corresponds to a probe volume with a length that is determined by the length of the emitted pulse and the window used for
signal processing. For the derivation of the method presented in this paper - which is based on the measured distance to the
sea surface - it is convenient to approximate the probe volume by a single point and, thus, a single measurement range. The



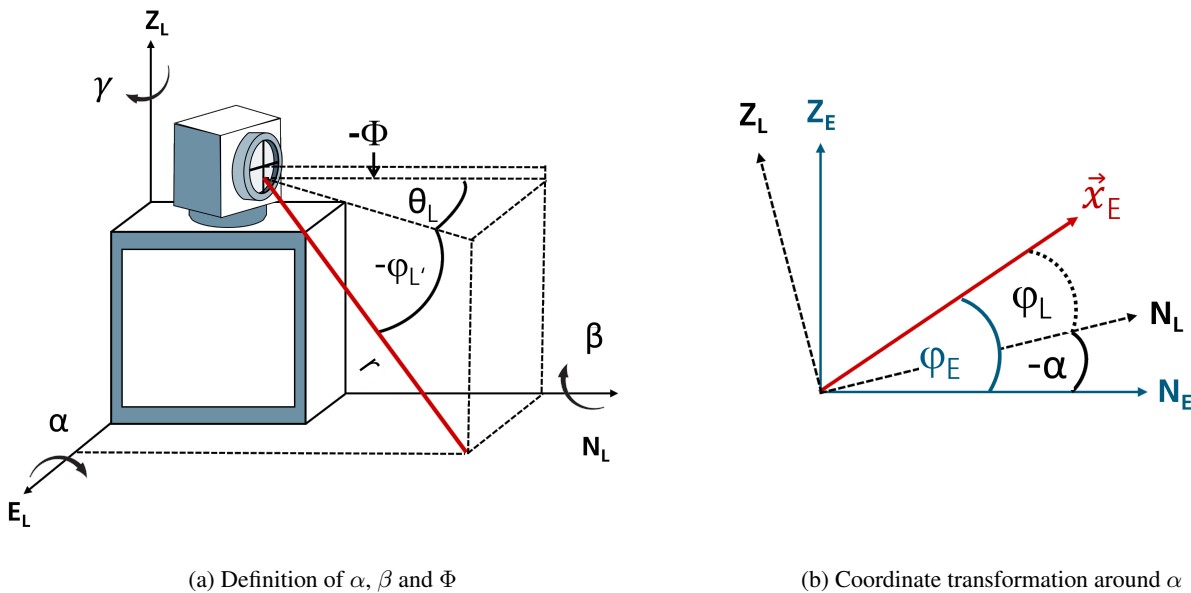

(a) Definition of $\alpha$, $\beta$ and $\Phi$    (b) Coordinate transformation around $\alpha$

**Figure 1.** (a) Illustration of a scanning lidar's orientation parameters. The laser beam (red) is defined by range $r$, elevation $\varphi_{L'}$, and azimuth $\theta_L$ in the lidar's internal coordinate system ($N_L$: device north, $E_L$: device east, $Z_L$: device vertical axis). Device tilt is given by pitch $\alpha$ (positive for downward tilt to the north), roll $\beta$ (positive for downward tilt to the west), and yaw $\gamma$ (positive clockwise). In addition, a negative elevation offset $\Phi$ is shown, shifting the laser beam downward. (b) 2D schematic of the transformation from the internal lidar coordinate system (L) to the external orthogonal (E). The external direction vector $\boldsymbol{x}_E$ (red) results from the lidar elevation $\varphi_L$ combined with pitch tilt.

motivation for this approximation and its implications are discussed in detail in Sect. 2.3, and a sensitivity analysis is presented in Sect. 3.1.1. The following equation defines the relationship between $\varphi_L$, $\theta_L$, $r$ and the lidar measurement location ($\boldsymbol{x}_L$):

$$\boldsymbol{x_L} = r \cdot \begin{pmatrix} \cos(\varphi_L) \cdot \sin(\theta_L) \\ \cos(\varphi_L) \cdot \cos(\theta_L) \\ \sin(\varphi_L) \end{pmatrix}. \tag{1}$$

The device's tilt is characterized by pitch $\alpha$ and roll $\beta$. Note that the definitions of pitch and roll vary across the literature.
Herein, pitch is defined as the rotation around the internal west-east axis of the device, whereby a positive pitch corresponds to a downward tilt in the direction of the device's north. Roll refers to rotation around the internal north-south axis, with a positive roll indicating a downward tilt toward the device's west (see Fig. 1a). Mathematically, the tilting by pitch and roll can be expressed by the following rotation matrices (e.g. Rott et al., 2022):



$$\mathbf{R_{pitch}} = \begin{pmatrix} 1 & 0 & 0 \\ 0 & \cos(\alpha) & \sin(\alpha) \\ 0 & -\sin(\alpha) & \cos(\alpha) \end{pmatrix} \tag{2}$$

$$\mathbf{R_{roll}} = \begin{pmatrix} \cos(\beta) & 0 & -\sin(\beta) \\ 0 & 1 & 0 \\ \sin(\beta) & 0 & \cos(\beta) \end{pmatrix}. \tag{3}$$

The rotation matrices are defined to describe the transformation from the tilted lidar coordinate system to the external orthogonal coordinate system. Figure 1b illustrates this rotation in two dimensions, using pitch as an example. The multiplication of the two matrices results in a matrix that describes the inclination of the lidar device in terms of pitch and roll:

$$\mathbf{R_{tilt}} = \mathbf{R_{pitch}} \cdot \mathbf{R_{roll}} = \begin{pmatrix} \cos(\beta) & 0 & -\sin(\beta) \\ \sin(\alpha) \cdot \sin(\beta) & \cos(\alpha) & \sin(\alpha) \cdot \cos(\beta) \\ \cos(\alpha) \cdot \sin(\beta) & -\sin(\alpha) & \cos(\alpha) \cdot \cos(\beta). \end{pmatrix} \tag{4}$$

Note that the order of rotations matters, thus $\mathbf{R_{pitch}} \cdot \mathbf{R_{roll}} \neq \mathbf{R_{roll}} \cdot \mathbf{R_{pitch}}$. The latter could also be a valid definition for $\mathbf{R_{tilt}}$. However, in the context of scanning lidar devices, pitch and roll are usually small enough such that equality between both possible definitions can be assumed.

To reach a specific point in the external coordinate system, defined by the external elevation angle $\varphi_E$, the corresponding internal elevation angle $\varphi_L$ must be determined. This requires accounting for the tilt of the device, as the internal and external coordinate systems are misaligned due to pitch and roll (see Fig. 1b).

However, besides tilt effects, vertical-axis deviations of the scanning mechanism (elevation offset) can also be relevant for alignment calibration. For the lidar considered here, Vaisala's WLS400S, a constant elevation offset $\Phi$ often occurs for technical reasons in the alignment of a scanner head (illustrated in Fig. 1a). The internal elevation angle $\varphi_L$ is composed of the programmable angle $\varphi_{L'}$ and an elevation offset $\Phi$:

$$\varphi_{\mathrm{L}} = \varphi_{\mathrm{L'}} + \Phi. \tag{5}$$

Here, $\varphi_{L'}$ is the angle defined within the lidar system, while $\varphi_L$ denotes the actual beam elevation, shifted by the offset and additionally affected by device tilt. A positive $\Phi$ corresponds to an upward shift relative to the ideal, offset-free orientation.

By applying Eqs. (1), (4), and (5), the lidar's internal direction vector $\boldsymbol{x_L}$ can be transformed into the corresponding external direction vector $\boldsymbol{x_E}$. This transformation is performed using a rotation matrix, assuming the coordinate origin is located at the laser beam exit point.





$$x_{\mathrm{E}} = \mathbf{R_{tilt}} \cdot x_{\mathrm{L}}$$

$$= r \cdot \begin{pmatrix} \cos(\beta) \cdot \cos(\varphi_{\mathrm{L}'} + \Phi) \cdot \sin(\theta_{\mathrm{L}}) - \sin(\beta) \cdot \sin(\varphi_{\mathrm{L}'} + \Phi) \\ \sin(\alpha) \cdot \sin(\beta) \cdot \cos(\varphi_{\mathrm{L}'} + \Phi) \cdot \sin(\theta_{\mathrm{L}}) + \cos(\alpha) \cdot \cos(\varphi_{\mathrm{L}'} + \Phi) \cdot \cos(\theta_{\mathrm{L}}) + \sin(\alpha) \cdot \cos(\beta) \cdot \sin(\varphi_{\mathrm{L}'} + \Phi) \\ \cos(\alpha) \cdot \sin(\beta) \cdot \cos(\varphi_{\mathrm{L}'} + \Phi) \cdot \sin(\theta_{\mathrm{L}}) - \sin(\alpha) \cdot \cos(\varphi_{\mathrm{L}'} + \Phi) \cdot \cos(\theta_{\mathrm{L}}) + \cos(\alpha) \cdot \cos(\beta) \cdot \sin(\varphi_{\mathrm{L}'} + \Phi) \end{pmatrix} \quad (6)$$

The azimuth angle $\theta_{\mathrm{L}}$ is corrected by applying a north offset $\gamma$, which accounts for the rotational misalignment of the lidar relative to true geographic north (see Fig. 1a):

$$\theta_{\mathrm{E}} = \theta_{\mathrm{L}} + \gamma. \quad (7)$$

$\theta_{\mathrm{E}}$ denotes the external azimuth angle in the orthogonal external coordinate system, and $\gamma$ is defined as positive when the lidar is rotated clockwise relative to true north. Assuming small pitch, roll and elevation angles $\theta_{\mathrm{L}} \approx \theta_{\mathrm{L}'}$, where $\theta_{\mathrm{L}'}$ represents the programmable internal azimuth angle set within the lidar system, excluding the mechanical offset in azimuth direction. In this study, the north offset is assumed to be known from independent measurements (e.g. CNR mapping of a hard target), and is therefore not considered in the following discussion.

For the SSL method presented herein, the third component (z-component) of Eq. (6) is evaluated to derive $\alpha$, $\beta$ and $\Phi$. The vertical (z-) component in the external coordinate system is given by the sine of the external elevation angle $\varphi_{\mathrm{E}}$:

$$\sin(\varphi_{\mathrm{E}}) = \frac{z}{r}. \quad (8)$$

From this, the following relationship is obtained:

$$\sin(\varphi_{\mathrm{E}}) = \cos(\alpha) \cdot \sin(\beta) \cdot \cos(\varphi_{\mathrm{L}'} + \Phi) \cdot \sin(\theta_{\mathrm{L}}) - \sin(\alpha) \cdot \cos(\varphi_{\mathrm{L}'} + \Phi) \cdot \cos(\theta_{\mathrm{L}}) + \cos(\alpha) \cdot \cos(\beta) \cdot \sin(\varphi_{\mathrm{L}'} + \Phi) \quad (9)$$

## 2.2 Generalisation of the SSL method

SSL is a method of determining the beam alignment calibration of a scanning lidar in offshore or nearshore environments. For this purpose, the sea surface is used as a target. In its original form proposed by Rott et al. (2022), the sea surface is considered to be a flat plane around the measuring device. If the laser beam of the instrument is directed at the water surface, the distance at which the laser beam hits the water surface will remain the same in all directions at a constant negative elevation angle in the case of ideal levelling. Deviations from this behaviour are caused by a tilt of the device and thus $\alpha$ and/or $\beta$ deviating from zero. (Waves may affect the uncertainty of the results, as discussed in Sect. 3.1.2).

On this basis, Rott et al. (2022) introduced the SSL method based on PPI scans. The values of pitch and roll were determined based on the directional differences in the distances between the lens of the laser beam and the entry point into the water surface



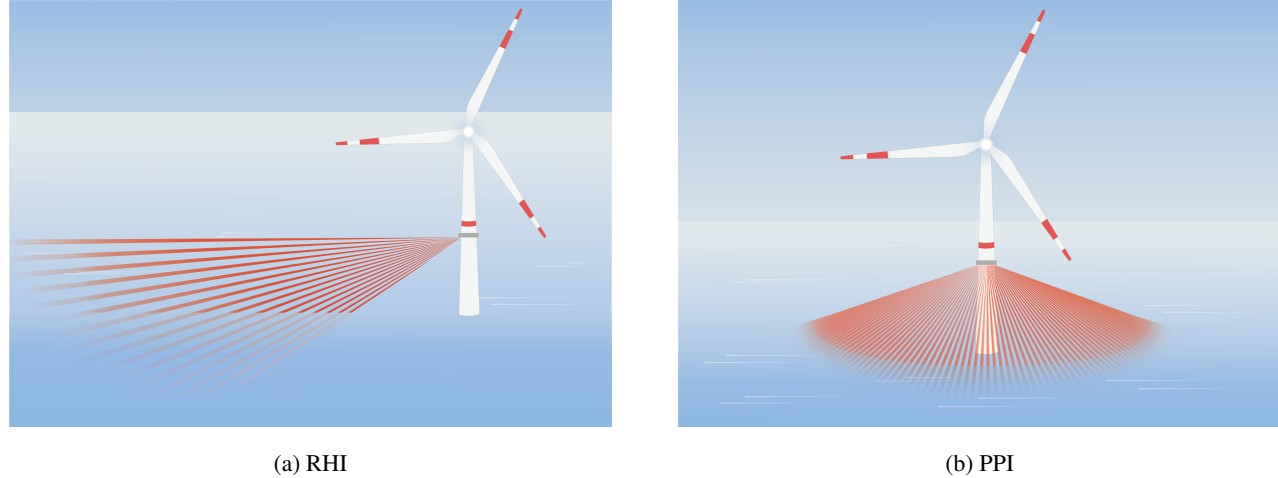

(a) RHI                                                                (b) PPI

**Figure 2.** Schematic representation of the SSL using (a) RHI and (b) PPI scan patterns. (a) shows an exemplary RHI at a single azimuth; for SSL, however, multiple azimuths (ideally covering $360°$) with elevation angles between $\varphi_{\text{start}}$ and $\varphi_{\text{end}}$ are required. (b) shows a PPI at a fixed elevation over a complete azimuthal sweep; for the extended SSL method, multiple PPI scans at different elevations within the same range are necessary. Images taken from Gramitzky et al. (2024).

at a fixed negative elevation angle. In addition, Champneys et al. (2024) and Gramitzky et al. (2024) introduced an alternative
procedure for applying the SSL method using RHI scans. In contrast to the PPI method, this technique does not analyse ranges at a fixed elevation angle. Instead, it determines the elevation angle for each azimuth direction at which a constant given distance to the water surface is reached. Figure 2 shows the two scan patterns schematically.

Both methods rely, in principle, on the relationship established in Eq. (9), utilizing the fact that

$$\sin(\varphi_{\text{E}}) = -\frac{h_{\text{lidar}}}{r_{\text{w}}}, \tag{10}$$

where $h_{\text{lidar}}$ denotes the height of the lidar above the sea surface and $r_{\text{w}}$ defines the distance to the sea surface where the laser beam dips into the water. As the elevation angle is defined as negative for downward directions, $h_{\text{lidar}}$ is treated as a positive quantity, and the negative sign in Eq. (10) ensures that $\varphi_{\text{E}}$ correctly represents a negative (downward) angle.

At distances of approximately $1.5$ km, it is important to take into account that the sea surface is not perfectly flat, but instead follows the curvature of the Earth. For instance, over a horizontal distance $d$ of $1.5$ km, this curvature causes a vertical drop of
roughly $0.2$ m relative to a horizontal reference plane, at $d = 2.5$ km, the drop increases to $0.5$ m, and at $d = 4$ km, it reaches $1.26$ m. This effect can be approximated by accounting for the height difference $z_{\text{earth}}$ due to Earth's curvature between the lidar position at $(d = 0)$ and a target point at a horizontal distance $(d = x)$ (Kahmen, 2011, pp. 456-459):

$$h_{\text{lidar,d=x}} = h_{\text{lidar,d=0}} - z_{\text{earth,d=x}} = h_{\text{lidar,d=0}} - \frac{x^2}{2 \cdot R}, \tag{11}$$





where $R$ denotes the Earth's radius (here, $R = 6371$ km) and $x$ is the horizontal distance from the lidar. Here, $h_{\mathrm{lidar,d=0}}$ repre-
sents the lidar height at the position of the lidar location $d = 0$. For simplicity, we will denote $h_{\mathrm{lidar,d=0}}$ as $h_{\mathrm{lidar}}$ in the follow-
ing. If $x \gg h_{\mathrm{lidar}}$, the path length can be approximated by the horizontal distance, i.e. $x \approx r$. For distances greater than $1$ km,
the difference between the actual path length and the horizontal distance becomes negligible, affecting the curvature-induced
height correction only at the third decimal place. Inserting Eq. (11) into Eqs. (9) and (10) and applying the approximation, the
resulting expression represents the extended SSL model including the effects of Earth's curvature and $\Phi$:

$$-\frac{h_{\mathrm{lidar}} - \frac{r_{\mathrm{w}}^2}{2 \cdot R}}{r_{\mathrm{w}}} = \cos(\alpha) \cdot \sin(\beta) \cdot \cos(\varphi_{\mathrm{L'}} + \Phi) \cdot \sin(\theta_{\mathrm{L}}) - \sin(\alpha) \cdot \cos(\varphi_{\mathrm{L'}} + \Phi) \cdot \cos(\theta_{\mathrm{L}}) + \cos(\alpha) \cdot \cos(\beta) \cdot \sin(\varphi_{\mathrm{L'}} + \Phi). \quad (12)$$

By acquiring measurements over multiple azimuth $\theta_{\mathrm{L}}$ and elevation $\varphi_{\mathrm{L'}}$ angles in combination with the corresponding
measurement distance $r_{\mathrm{w}}$, the system parameters $h_{\mathrm{lidar}}$, $\alpha$, $\beta$ and $\Phi$ can be inferred from Eq. (12). A suitable approach for this
estimation is, for example, the application of a least-squares fitting method. Rott et al. (2022) use PPI scans for this purpose
and simplify Eq. (12) by assuming $\Phi = 0$ and neglecting $h_{\mathrm{earth}}$. In contrast, our approach enables direct estimation of $\Phi$ within
the fitting process, removing the need for such assumptions.

Making several approximations for small angles, Eq. (12) can be simplified. In most practical applications, $\alpha$ and $\beta$ do
not exceed $0.2°$. Thus $\cos(\alpha) \approx \cos(\beta) \approx 1$, $\sin(\alpha) \approx \alpha$ and $\sin(\beta) \approx \beta$ provide a highly accurate approximation. Scanning
the sea surface at larger distances, $\varphi_{\mathrm{L'}}$ becomes small and $\Phi$ is generally small for properly manufactured scanning lidars.
Applying the approximations $\sin(\varphi_{\mathrm{L'}} + \Phi) \approx \varphi_{\mathrm{L'}} + \Phi$ and $\cos(\varphi_{\mathrm{L'}} + \Phi) \approx 1$ and rearranging Eq. (12) for $\varphi_{\mathrm{L'}}$, the resulting
expression represents a simplified equation of the extended SSL method:

$$\varphi_{\mathrm{L'}} = \alpha \cdot \cos(\theta_{\mathrm{L}}) - \beta \cdot \sin(\theta_{\mathrm{L}}) - \frac{h_{\mathrm{lidar}} - \frac{r_{\mathrm{w}}^2}{2 \cdot R}}{r_{\mathrm{w}}} - \Phi. \quad (13)$$

The accuracy of this simplification is verified by comparing Eq. (12) with the simplified Eq. (13). Even in extreme cases,
with $\alpha$, $\beta$ and $\Phi$ up to $1°$, the deviations at $\varphi = -3.5°$ are less than $0.01°$. Under typical conditions, with $\alpha, \beta, \Phi \leq 0.2°$, the
deviation at $\varphi = -3.5°$ is less than $0.003°$, and therefore negligible. For steeper elevation angles, for example at $\varphi = -7°$,
the deviation is less than $0.02°$, but increases to about $0.06°$ at $\varphi = -10°$. This demonstrates that the simplified formulation
(Eq. (13)) is very accurate for small elevation angles, but its validity is limited for larger elevation angles. In this case, Eq. (12)
should be applied.

In general, the extended SSL method is not limited to a specific scan type such as PPI or RHI. It is important that distance
measurements to the water surface are available for different azimuth and elevation angles. With multiple RHI scans, one
automatically obtains measurements for various elevation angles. In the case of PPI, at least two different elevation angles
must be considered. The difference is that in RHI scanning, the beam is continuously moved through elevation angles, and
the returned signal is an integration between two elevation steps. Therefore a small angular resolution should be chosen. In
contrast, PPI scanning is performed at discrete elevation angles in step-stare mode.





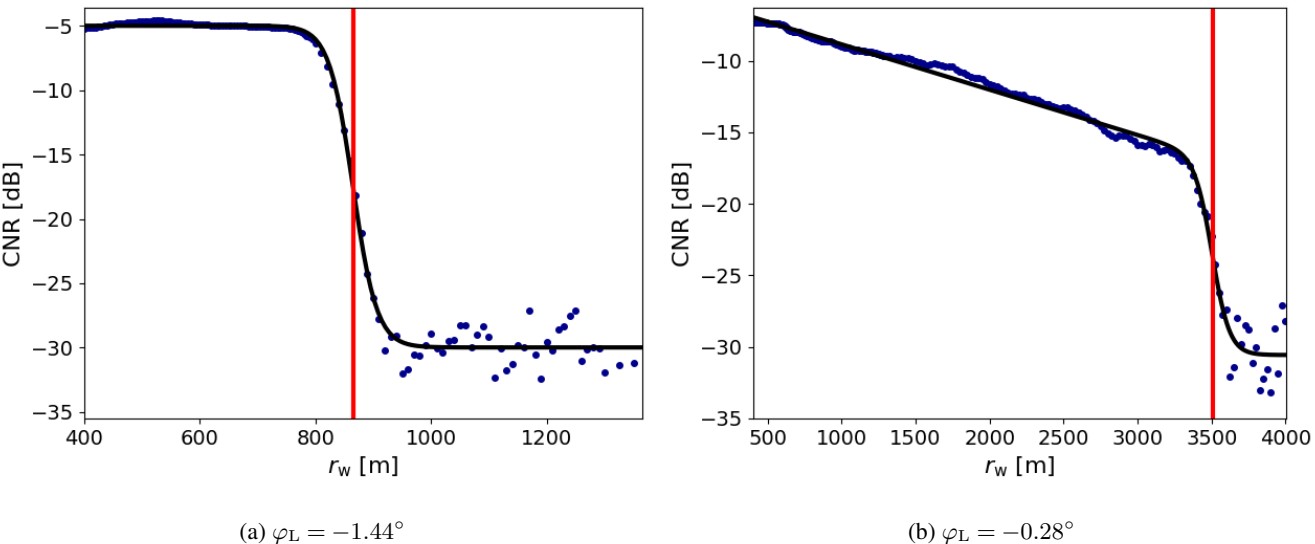

(a) $\varphi_\mathrm{L} = -1.44°$  (b) $\varphi_\mathrm{L} = -0.28°$

**Figure 3.** CNR signal versus $r_\mathrm{w}$ for (a) steeper elevation $\varphi_L = -1.44°$ and (b) flatter elevation $\varphi_L = -0.28°$. Blue dots show measurements; the black line is the fitted model. The red line marks the inflection point $i_\mathrm{p}$, interpreted as the estimated water entry point $r_\mathrm{w}$. In (a), the CNR drop occurs at the water entry point, allowing us to assume $A = 0$, whereas in (b), a continuous decrease in CNR with distance is observed, requiring $A < 0$. The data shown were recorded using an RHI scan

## 2.3 CNR-based range detection

The SSL method requires an accurate estimation of the range $r_w$ between the laser emission point of a scanning lidar and the the water surface at a fixed elevation and azimuth angle. For SSL, $r_w$ is derived from the evolution of CNR over the range. If the laser beam of a lidar hits a solid object, in general, a very strong back-scatter signal is obtained. However, in contrast, when the beam hits the sea surface, the signal usually exhibits a sharp decrease in CNR values around this distance (Rott et al., 2022). This behaviour, seen in Fig. 3, can be attributed to the ability of the water surface to absorb parts of the laser light

and/or reflect it in directions other than back to the lidar device.

Rott et al. (2022) suggested using the inflection point of the sharp drop-off of the (logarithmic) CNR signal over distance for determining the immersion point of the laser beam into the water, as seen in Fig. 3. For this purpose, they fitted an inverse sigmoid function to the observed CNR signal. The inflection point of this function is then considered to be the point at which the laser beam enters the water surface. A shortcoming of their method is that at larger distances (flat elevation angles), the

230 typical decrease of CNR values within the atmosphere for increasing distance is not taken into account. Gramitzky et al. (2024) suggested accounting for this effect by introducing a linear term $(1 + A \cdot (r - i_\mathrm{p}))$ into the sigmoid function. The modified equation for determining the distance, used herein, is:





$$CNR(r) = \frac{(CNR_{\text{high}} - CNR_{\text{low}}) \cdot (1 + A \cdot (r - i_{\text{p}}))}{1 + \exp((r - i_{\text{p}}) \cdot g)} + CNR_{\text{low}}, \tag{14}$$

where $CNR_{\text{high}}$ and $CNR_{\text{low}}$ are the maximum and minimum values of the inverse sigmoid function, $A$ is a constant that describes the shape of the CNR signal over distance before the beam hits the water and $g$ is the growth rate of the function that describes the rate of decrease of the signal strength around the immersion point $i_{\text{p}}$. Note that, due to the additional term, the inflection point is not exactly equal to $i_{\text{p}}$. However, the difference is very small and for the further discussion we will use the term inflection point interchangeably to $i_{\text{p}}$. $A$, $i_{\text{p}}$, $g$, $CNR_{\text{high}}$ and $CNR_{\text{low}}$ are derived by fitting Eq. (14) to the observations of the CNR signal data. In this analysis, the limits for $A$ were defined as $-0.01 \text{ m}^{-1} \leq A \leq 0 \text{ m}^{-1}$. These limits were derived from inspection of the derived ranges to exclude outliers and are not physically motivated. $g$ was constrained to lie between 0 and 1, and later filtered according to the criteria described in Sect. 2.6. Additionally, limits for $i_{\text{p}}$, as well as the $CNR_{\text{high}}$ and $CNR_{\text{low}}$, were applied depending on the specific settings of the RHI or PPI scans.

While the inflection point provides a visually intuitive location for the distance to the water surface, its location does not directly correspond to the distance between the lidar and the water surface. Pulsed lidars average the signal over a certain distance - often called probe length - that is defined by the length and shape of the laser pulse and the internal signal processing (e.g. Pauscher et al., 2016) of the lidar. When probing the atmosphere, the distance measure provided by the lidar should ideally correspond to the distance where approx. half of the signal is returning from before and half of the signal is returning from behind that distance. If the weighting function along the beam of the lidar is assumed to be symmetric, this corresponds to the middle of probe volume.

In contrast, due to the logarithmic scale of the CNR, the inflection point of the steep CNR decline corresponds to a distance with a signal several orders of magnitude smaller than at ranges before entering the water. This means that almost all of the probe volume is inside the water at the distance of the location of the inflection point. As a consequence, the distance of the inflection point needs to be corrected (reduced) to match the distance to the water surface.

In theory, the distance of the inflection point could be modelled by integrating the weighting function of the probe volume along the beam. The signal strength with the presence of a water surface ($\hat{P}$) at the distance $b_w$ as a function of the signal strength without the presence of a water surface ($P$) can be written as (e.g. Mann et al., 2009):

$$\hat{P}(r) = \int\limits_{-\inf}^{b_w} \chi(s + r) P(r) ds, \tag{15}$$

where $\chi(s)$ is the weighting function of the signal along the beam. The exact modelling of the CNR drop-off with increasing $r$ requires detailed knowledge of $\chi(s)$. While several studies have used models to describe the weighting inside the probe volume to e.g. model turbulence attenuation (Mann et al., 2009; Pauscher et al., 2016; Peña and Mann, 2019), the exact shape of $\chi(s)$ remains uncertain and depends on both pulse shape and signal processing and therefore also on the device type (e.g. Pauscher et al., 2016). Detailed information about these aspects was not available to the authors. Moreover, at the inflection point, as well



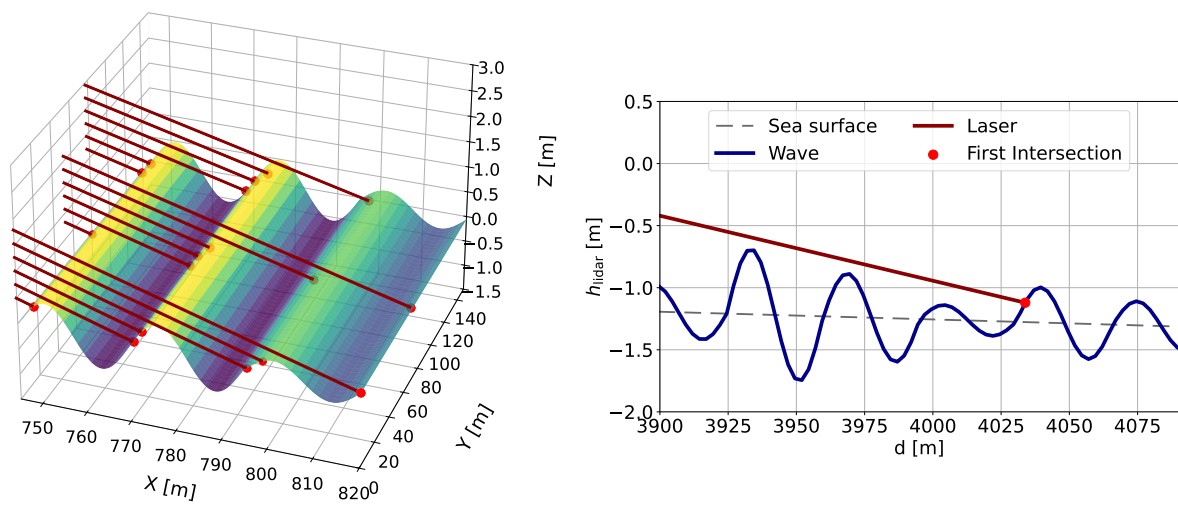

(a) 3D laser–wave interaction, varying $\varphi$ and $\theta$          (b) 2D laser–wave interaction along laser beam

**Figure 4.** Schematic representation of the laser's behaviour on a wave-covered water surface. Waves follow a Rayleigh distribution and propagate along the x-axis. (a) shows the same elevation angles for three azimuth directions; red dots mark intersections with the water surface. The beam's distance to the wave varies with azimuth, depending on whether it hits a crest. (b) highlights this effect at $\theta = 45°$; the dashed line indicates Earth's curvature relative to the lidar's sea surface height.

for the majority of the steep CNR drop-off, the signal is several orders of magnitude smaller than the signal before the probe volume enters the water (see Fig. 3). This makes modelling of the CNR drop-off extremely sensitive to assumptions about the

tail end of $\chi(s)$ - an area where models of $\chi(s)$ are likely to be very uncertain.

     For the reasons outlined above, rather than modelling the return signal as a function of range, a simple constant distance correction is applied in this paper. In the presented results half the probe length $r_{\mathrm{w}} = i_{\mathrm{p}} - 37.5$ m is chosen (probe volume length: 75 m). This choice is motivated by applying several simple assumption about the change of the return signal with $r$. Firstly, we assume that $r$ as provided by the lidar, corresponds to be in the middle of the probe volume. Secondly, it is assumed

that, outside of the probe volume length provided by the manufacturer, the share of the return signal has reduced several orders of magnitude (to the inflection point of the CNR-over-range-curve) and (almost) no signal is returned form the distances closer than the water surface. We acknowledge that this is only a relatively rough approximation and therefore perform a sensitivity analysis to ranging uncertainties in Sect. 3.1.1.





## 2.4 Simulation of the sea surface with waves

An important assumption that is made in Eqs. (12) and (13) is that the sea surface can be approximated by a flat surface. However, due to the presence of surface waves, this assumption is never fully satisfied in real-world conditions. Consequently, it is commonly advised to perform a SSL under calm wind conditions with minimal wave activity (Rott et al., 2022; Gramitzky et al., 2024). However, the sea surface will never be entirely flat. In this analysis, we therefore assess the errors/uncertainties introduced into SSL by waves of different amplitudes and wavelengths. The characteristics of water waves depend on a va-

riety of physical factors, e.g., wind speed, water depth, currents (Holthuijsen, 2010). It is not possible to cover all conditions completely here. Therefore, a simplified and idealised simulation approach is chosen to quantify the influence of basic wave parameters on measurement uncertainty of SSL.

While the SSL equation assumes that the surface is influenced exclusively by the curvature of the Earth (see Sect. 2.2), a wave-shaped altitude profile of the sea surface is now introduced. his model considers a sinusoidal wave with varying ampli-

tudes $A$, propagating along the x-direction and characterized by a defined wavelength $\lambda$. For the analysis of wave effects, note that the x-direction is arbitrary and does not affect the results because the lidar scan covers the full $360°$ circle. The water surface is divided into segments of length $\lambda$, with each segment $i$ assigned an amplitude $A_i$. The water surface is divided into segments of length $\lambda$, with each segment $i$ being assigned an amplitude $A_i$. The resulting surface height $z_{sur}$ is obtained from the superposition of the wave and the curvature term of the Earth:

$$z_{sur}(x,y) = A_i \cdot \sin\left(\frac{2\pi}{\lambda}(x+x_0)\right) - \frac{x^2 + y^2}{2 \cdot R}. \tag{16}$$

Here, $x_0 = v \cdot t$ describes a displacement of the wave along the x-direction where $t$ is time and $v$ wave velocity. The wave velocity is determined according to the linear dispersion relation for surface gravity waves in finite water depth (Holthuijsen, 2010, p. 125):

$$v = \sqrt{\frac{g_e \lambda}{2\pi} \tanh\left(\frac{2\pi s}{\lambda}\right)}, \tag{17}$$

where $g_e$ is the gravitational acceleration and $s$ the water depth.

For the amplitudes $A_i$ of the individual wave segments, a Rayleigh distribution is assumed, which is often used to describe the statistical distribution of wave heights in wind-driven seas (Holthuijsen, 2010, pp.33–36). Assuming that the waves are predominantly regular, sinusoidal shapes with random superposition, a relationship between the scale parameter ($\sigma$) of the Rayleigh distribution and the significant wave height ($H_s$) can be approximated as $H_s \approx 4 \cdot \sigma$ (Vinje, 1989). Here, the significant

wave height $H_s$, defined as the arithmetic mean of the upper third of wave heights within an observation period, provides a statistical measure that directly determines the scale parameter $\sigma$ of the Rayleigh distribution. In the simulation, specific amplitudes $A_i$ are drawn for each wavelength segment $i$ from the Rayleigh distribution with the scale parameter $\sigma$.

The amplitude distribution remains constant across the entire domain, while the wave shifts over time via the phase parameter $x_0$. This generates realistic, non-stationary waves with different amplitudes, which are shifted in relation to the position of the





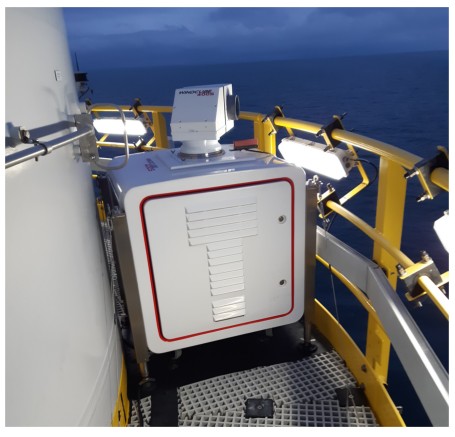

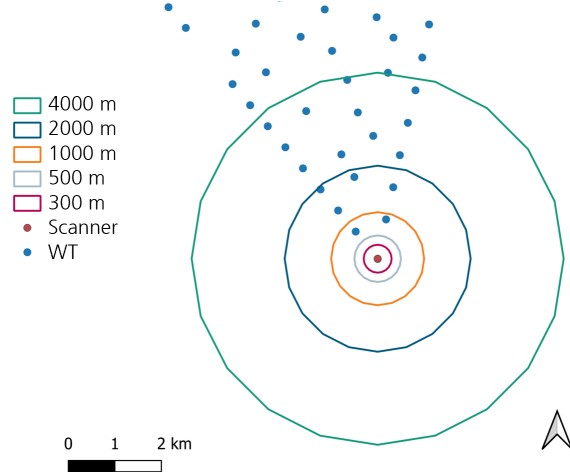

(a) Scanning Lidar WLS400S ©Jens Riechert

(b) Scanning Lidar position in the wind farm layout

**Figure 5.** (a) shows the scanning lidar installed on the transition piece of a wind turbine, (b) displays the wind farm layout, with blue dots indicating turbine positions and a red dot marking the lidar location. The circles represent fixed radii around the lidar, which approximately reflect the distances assuming a horizontal device alignment.

lidar. Figure 4a and Fig. 4b show examples of the simulated wave model for selected elevation and azimuth angles and a specific significant wave height. The first intersection point between the laser beam and the wave is determined (red points) and the distance is calculated. A wave movement is simulated by changing $x_0$ accounting for the time required by the lidar between to change between beam directions.

## 2.5   Measurement campaign

The experimental results in this study are based on an offshore measurement campaign conducted in the North Sea using a WindCube WLS400S (Vaisala) long-range scanning lidar installed on a transition piece of a wind turbine (see Fig. 5a). Figure 5b illustrates the measurement setup of the wind farm, indicating the location of the scanning lidar at the southern edge of the wind farm. The circles represent example distances, showing that other wind turbines (blue dots) within the farm fall within the laser beam's line of sight at various ranges for specific azimuth angles. The SSL measurements were carried out

during periods with wind speeds $< 4$ m/s, from 19 November 2023, 22:00:00 UTC to 20 November 2023, 08:00:00 UTC, ensuring that the wind turbine was not running. This minimizes the effects of platform inclination caused by the thrust of the wind turbine during the measurement. For this date, the sea state charts from the "Federal Maritime and Hydrographic Agency (BSH)" indicate significant wave heights of up to $1$ m (Deutscher Wetterdienst, 2023). These wave heights correspond to a wavelength of approximately $25$ m, which corresponds to a typical wave steepness of $0.04$ for deep water (Heineke and

Verhagen, 2009).



The settings used for the measurements were as follows: Two sequences of SSLs were carried out using the RHI method (see Fig. 2a). The RHI scans of the first sequence (RHI 1) ranged from start elevation angle $\varphi_{\text{start}}$ $-1.5°$ to end elevation angle $\varphi_{\text{end}}$ $-0°$ and the second sequence (RHI 2) from $-3°$ to $-1.5°$, both with a step size of $0.02°$. The azimuth resolution for both scans was $5°$. This sequence was followed by PPI scans (see Fig. 2b) with elevation angles of $-3°$ to $-0.5°$ in $0.5°$ steps. The resolution in the azimuth angle was $1°$. A full run of the RHI routine required approximately 30 min. The six PPI scans used to perform a PPI-based SSL took approximately 25 min. During the observation period, this allowed for the acquisition of 7-8 complete SSL datasets per measurement configuration.

All scans were performed with an accumulation time of 500 ms and a pulse length of 75 m. The resolution of the distance was adjusted depending on the measurement setting and the maximum distance, as only a maximum number of 319 range gates was possible. In order to eliminate any backlash effects, a single Line-of-Sight (LOS) scan located just below the measurement setting was performed before each PPI and RHI scan so that each scan was always initiated from the same direction. Due to the wind turbine tower blocking the line of sight, no measurements could be obtained in the azimuth range between $45°$ and $180°$. For comparison, 10-minute mean values from an inclinometer installed inside the wind turbine at the platform level were used. A hard target calibration carried out by the external consultancy DNV (2022) was used to validate the estimation of $\Phi$.

## 2.6 Data filtering

To minimise the impact of inaccurate distance estimations from the CNR-over-range signal, automated and targeted data filtering was applied based on the following criteria:

1. *Initial value filter* Measurements with an initial CNR value below -21 dB were excluded, as it is assumed that the laser beam was already obstructed in the first few meters.

2. *CNR-Filter* All measurement points with maximum CNR values above 0 dB were excluded, as this indicates hard reflective targets (e.g., neighbouring wind turbines, ships, or birds) that can distort the distance measurement.

3. *Growth rate filter* To identify CNR curves where the turning point cannot be reliably determined by the optimised sigmoid function, a filter was applied to the growth rate of the function. A very high or very low growth rate can indicate an unreliable fit of the data to Eq. (14). For this analysis we removed all measurement with a growth rate larger than 0.07 and smaller than 0.007. These values showed to be reliable in removing outliers in the range estimates from the CNR signal. This range should be adjusted depending on meteorological conditions.

4. *Manual filter* In a final step, outliers were manually screened and removed if an atypical CNR-over-range behaviour was observed. In the present measurement, this affected approximately 5–10 discrete azimuth angles, depending on the setting.



# 3 Results

The following section first presents a sensitivity analysis of relevant uncertainty parameters for SSL (Sect. 3.1), followed by an experimental evaluation of the SSL performance using data collected during an offshore measurement campaign in the North Sea using different parameter settings (Sect. 3.2).

## 3.1 Sensitivity analysis and parameter choice for SSL

The accuracy of the SSL results for pitch $\alpha$, roll $\beta$, and elevation offset $\Phi$ (Sect. 2.2; Eq. (13)) strongly depends on the assumptions made about the range of the laser's entry point into the sea surface $r_\mathrm{w}$ (Sect. 2.3) and a good approximation of the sea surface, which is influenced by wave-induced effects. The following sections present a sensitivity analysis of these factors.

### 3.1.1 Influence of range error on the uncertainty of the SSL results

In the SSL method, the distance to the water surface $r_\mathrm{w}$ is derived from the CNR signal as a function of range. As discussed in Sect. 2.3, determining the exact point at which the laser beam enters the water surface is subject to a certain degree of uncertainty, which in turn has an effect on the results of the SSL. The following analysis investigates the impact of a constant systematic error $\epsilon_\mathrm{r}$ in the determination of the distance to the water surface on the resulting measurements.

For this purpose, theoretical $r_\mathrm{w}$ are simulated for different azimuth and elevation angles. These simulated $r_\mathrm{w}$ values represent the measured $r_\mathrm{w}$ values in an SSL measurement and are computed using Eq. (13) with predefined parameters for pitch $\alpha$, roll $\beta$, elevation offset $\Phi$, and lidar height $h_\mathrm{lidar}$. $\epsilon_r$ is then added to the simulated $r_\mathrm{w}$ values. To estimate the SSL parameters, a least-squares adjustment method is applied, which evaluates the data using Eq. (13) with and without $\epsilon_r$. Finally, the deviations caused by $\epsilon_r$ are computed.

The following analysis investigates the impact of varying elevation interval boundaries for $\varphi_\mathrm{start}$ and $\varphi_\mathrm{end}$ on the resulting measurements. Figure 6a illustrates the effect for $\Delta\Phi$ with a constant $\epsilon_\mathrm{r}$ of $-37.5$ m (corresponding to half the pulse length of the laser beam; see Sect. 2.3). $\alpha$, $\beta$ and $\Phi$ are set to zero, so that only the influence of $\epsilon_\mathrm{r}$ is considered as the dominant factor. Furthermore, a $h_\mathrm{lidar}$ of 20 m is assumed, which influences the distance of $r_\mathrm{w}$ depending on the elevation angle. All intermediate elevation values are included in the evaluation with a step size of $0.02°$. The azimuth values range from $0°$ to $360°$ in $5°$ increments. In general, due to the reduced measured distance (negative $\epsilon_\mathrm{r}$), in Fig. 6a a positive deviation in $\Delta\Phi$ can be observed. While this may appear counter-intuitive, given that a shorter measured $r_\mathrm{w}$ suggests the lidar is closer to the sea surface, this is caused by the simultaneous error in $h_\mathrm{lidar}$, which is negative (see Fig. 7b).

As the elevation interval increases, the relative impact of $\epsilon_\mathrm{r}$ decreases, reducing its influence on the estimation of $\Phi$. For an elevation interval from $-3°$ to $-1.5°$ (comparable to the interval used in RHI 2), a error in $r_\mathrm{w}$ of $-37.5$ m causes a substantial $\Delta\Phi$ of approximately $-0.16°$. In contrast, within the elevation range from $-1.5°$ to $-0.3°$ (comparable to RHI 1), the same $\epsilon_\mathrm{r}$ results in deviations in $\Delta\Phi$ below $-0.02°$, indicating a significantly smaller impact. $\Delta\Phi$ becomes negligible when considering only very shallow intervals, such as between $-0.5°$ to $0°$. At these small angles, the lidar height of 20 m corresponds to



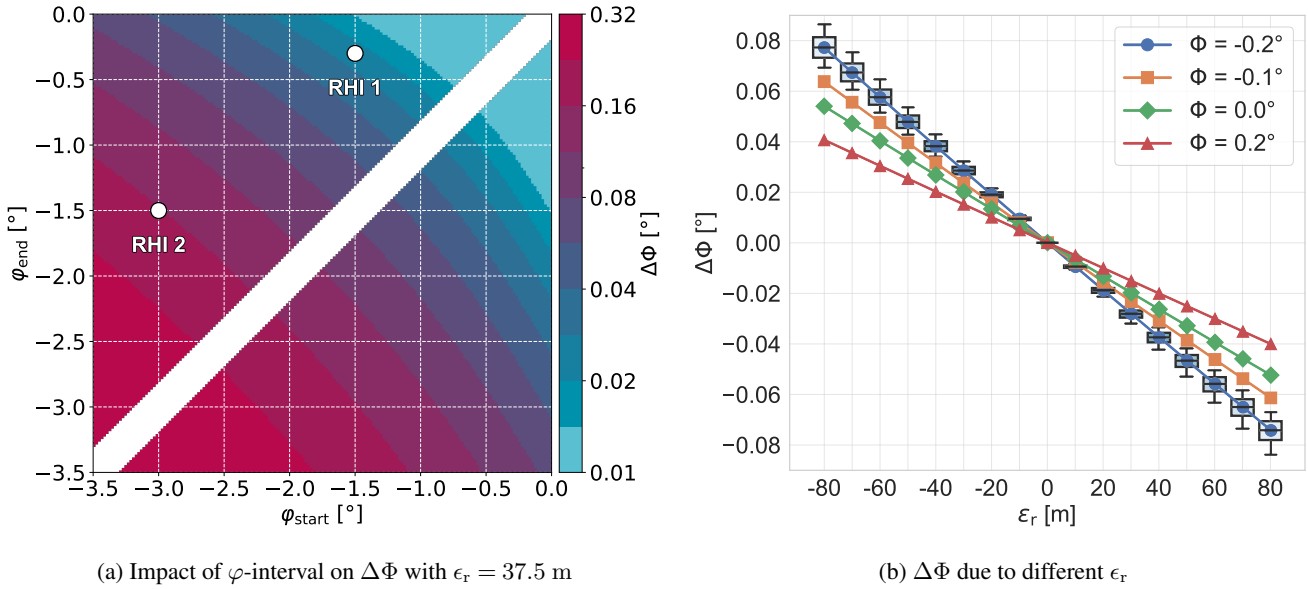

(a) Impact of $\varphi$-interval on $\Delta\Phi$ with $\epsilon_\mathrm{r} = 37.5$ m

(b) $\Delta\Phi$ due to different $\epsilon_\mathrm{r}$

**Figure 6.** Modelling results of the elevation offset $\Delta\Phi$ depending on $\epsilon_\mathrm{r}$ based on RHI scan settings (step size of $0.02°$ for $\varphi$ and $5°$ for $\theta$, with $h_\mathrm{lidar} = 20$ m). (a) shows the impact of the elevation angle interval on $\Delta\Phi$ for $\epsilon_\mathrm{r} = 37.5$ m. $\varphi_\mathrm{start}$ and $\varphi_\mathrm{end}$ are the start and end values for the interval of $\varphi$ used in the SSL evaluation. White areas denote NaN values, where the interval between $\varphi_\mathrm{start}$ and $\varphi_\mathrm{end}$ is too small to yield meaningful results. (b) shows the median value of $\Delta\Phi$ from the simulation due to varying $\epsilon_\mathrm{r}$ for different $\Phi$ (different coloured lines) using a fixed interval from $\varphi_\mathrm{start} = -1.5°$ to $\varphi_\mathrm{end} = -0.3°$ (RHI 1 settings). Box plots show the effect on $\Delta\Phi$ for varying $\alpha, \beta$ by $\pm 0.2°$ and $h_\mathrm{lidar}$ by 18–22 m for $\Phi = -0.2°$.

$r_\mathrm{w} \approx 3820$m at $\varphi = -0.3°$. Here, a change of the elevation angle of $0.02°$ corresponds a to a change of roughly $\pm 200$m in $r_\mathrm{w}$. Thus, $\epsilon_\mathrm{r}$ becomes small relative to the variation in $r_\mathrm{w}$, and its influence on overall accuracy is minimal.

However, at flat elevation angles the main challenge is the limited range of the lidar, at which the CNR drop-off can be reliably identified before the CNR-signal reaches the noise floor. With increasing distance, the CNR signal can become signif-
icantly weaker, making it difficult to accurately determine $r_\mathrm{w}$, since the characteristic CNR drop-off manifests only within a narrow dB interval. In Fig. 3b $r_\mathrm{w}$ can still be determined, although a decline in CNR is already apparent; at flatter elevation angles, the decline intensifies and the CNR drop interval decreases. Nevertheless, despite the presence of $\epsilon_\mathrm{r}$, the root mean square error (RMSE) of the model fit remains below $0.03°$ for almost all elevation interval combinations, with the majority even below $0.01°$ (not shown in the Figures).

To evaluate the effect of different $\epsilon_\mathrm{r}$ on the estimation of $\Phi$, a fixed elevation interval between $-1.5°$ and $-0.3°$ (RHI 1 settings) is considered (Fig. 6a). $\epsilon_\mathrm{r}$ is applied to all $r_\mathrm{w}$ simulated within this range, with typical variations in pitch and roll ($\pm 0.2°$) and a $h_\mathrm{lidar}$ difference due to tide (18–22 m). Distances beyond $4000$ m are excluded, since in the experimental data $r_\mathrm{w}$ could no longer be reliably detected at greater ranges. To ensure consistency between experimental data and model, the same distance threshold was applied in the simulation. As shown in Fig. 6a, $\Delta\Phi$ is positive for negative $\epsilon_\mathrm{r}$, indicating an




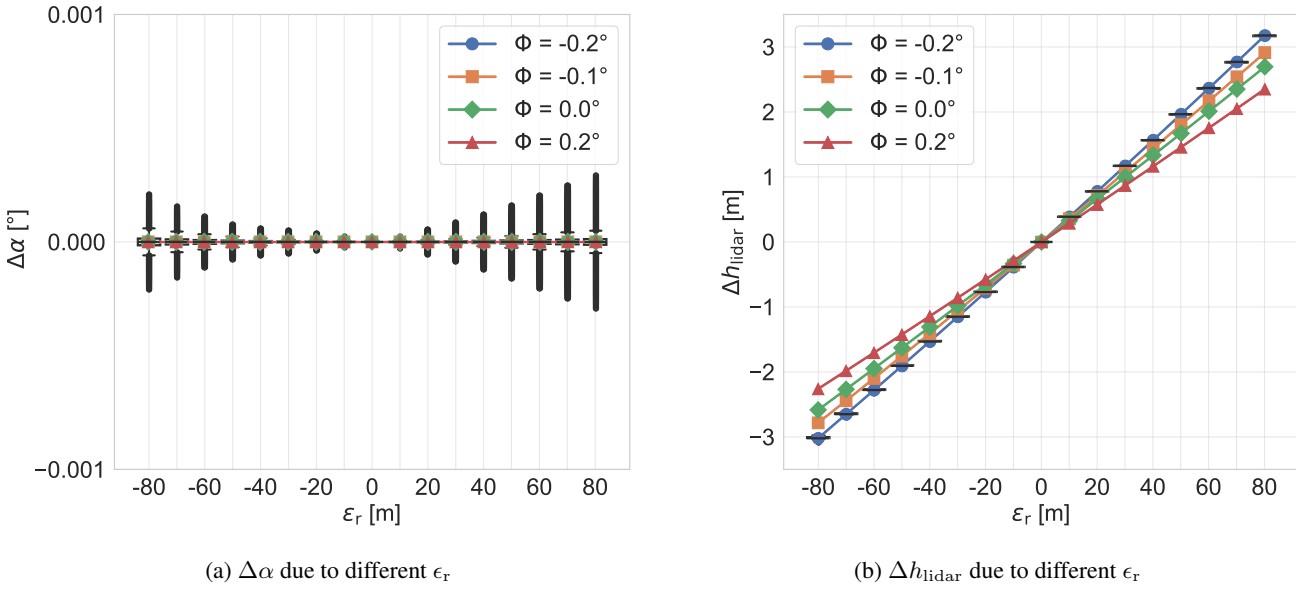

(a) $\Delta\alpha$ due to different $\epsilon_\mathrm{r}$      (b) $\Delta h_\mathrm{lidar}$ due to different $\epsilon_\mathrm{r}$

**Figure 7.** Simulated impact of different $\epsilon_\mathrm{r}$ (x-axis), (a) on $\alpha$ and (b) on $h_\mathrm{lidar}$. The figures show results for RHI 1 $\varphi$ settings. The box plots show the distribution for $\Phi = -0.2°$, but variations of $\alpha$ and $\beta$ between $-0.2°$ and $0.2°$ and $h_\mathrm{lidar}$ between $18$ and $22$ m. The lines show the medians of the results for different $\Phi$, similar to Fig. 6.

overestimation of $\Phi$ due to the $\epsilon_\mathrm{r}$. Conversely, for cases with positive $\epsilon_\mathrm{r}$, $\Delta\Phi$ becomes negative. $\Delta\Phi$ decreases almost linearly with distance deviations from $-80$ m to $+80$ m, with a median $\Delta\Phi$ of $0.08°$ at $-80$ m and approximately $0.04°$ at $-40$ m for $\Phi = 0.2°$. Among influencing factors, the simulated $\Phi$ has the most significant impact on $\Delta\Phi$, whereas pitch, roll, and $h_\mathrm{lidar}$ contribute marginally ($< 0.01°$). For example, a $\epsilon_\mathrm{r}$ of $-80$ m leads to $\Delta\Phi \approx 0.04°$ for $\Phi = 0.2°$ (red line), nearly half the value observed for $\Phi = -0.2°$ under identical conditions (blue line).

Unlike the results for $\Delta\Phi$, Fig. 7a shows that simulated $\epsilon_\mathrm{r}$ have minimal impact on pitch or roll angle deviations (results of roll angle not shown due to their similarity). The simulation clearly indicates that uncertainties in pitch induced by $\epsilon_\mathrm{r}$ remain well below $0.001°$ and can therefore be considered negligible. However, $\epsilon_\mathrm{r}$ has a noticeable influence on the derived $h_\mathrm{lidar}$ (Fig. 7b), which can lead to deviations of up to $\pm 3$ m for $\epsilon_\mathrm{r}$ of $\pm 80$ m. The lidar height above sea level is generally not the focus of the SSL analysis, but it must be determined as an indirect variable in Eq. (13).

**3.1.2   Uncertainty due to waves**

For the SSL approach, Eq. (13) (Sect. 2.2) models the water surface as a plane shaped only by Earth's curvature. Therefore, measurements are ideally conducted with low wave amplitudes to reduce perturbations. However, in open-sea conditions, waves are always present. The following analysis simulates the impact of these wave conditions on SSL measurement accuracy. For this purpose, the wave model introduced in Sect. 2.4 is employed to calculate the difference $r_\mathrm{w}$ between the flat water surface




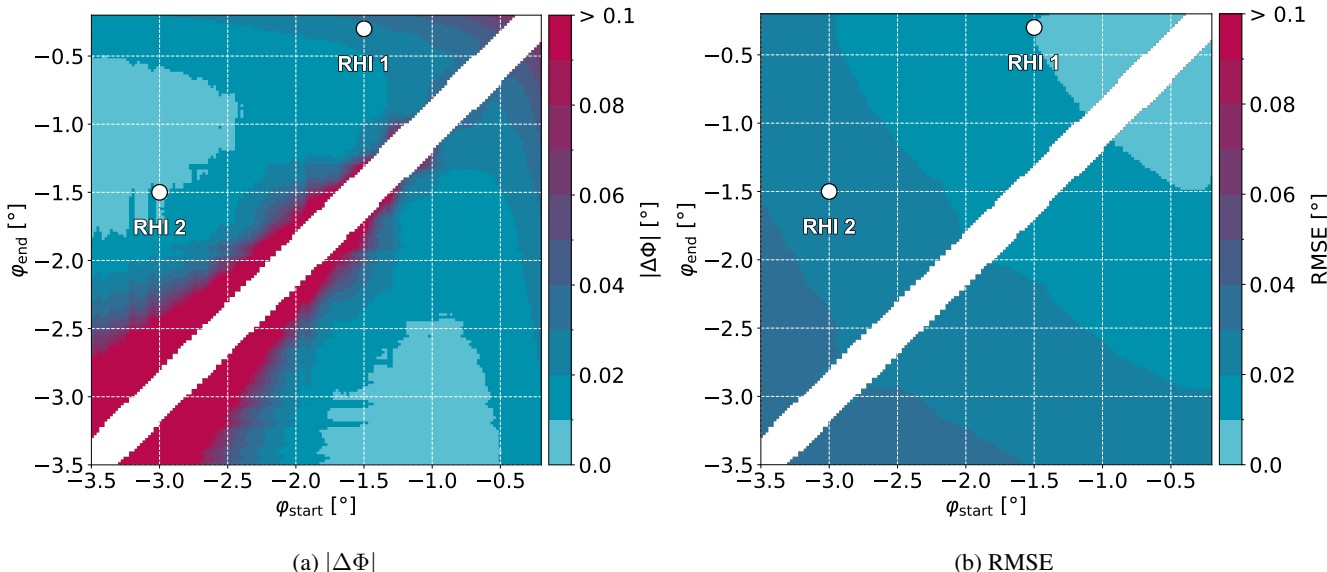

(a) $|\Delta\Phi|$          (b) RMSE

**Figure 8.** Influence of waves on $\Delta\Phi$, determined via SSL as a function of the elevation interval based on RHI settings, azimuth angle steps of $10°$, and elevation interval steps of $0.02°$. Waves with $H_s = 1$ m and $\lambda = 25$ m are considered. (a) shows the magnitude of $\Delta\Phi$ resulting from the influence of the simulated waves, while (b) shows the influence of waves for the RMSE of $\varphi_L$ in the model fit.

assumption and the intersection with the modelled wave profile. For this scenario, the lidar is again assumed to be perfectly aligned, i.e. $\alpha$, $\beta$, and $\Phi$ are set to zero.

     The analysis shows that, under wave conditions comparable to those during the measurement campaign, wave-induced effects on $\Delta\Phi$ are generally small, but specific narrow elevation intervals exhibit significant deviations, highlighting configurations that should be avoided. Figure 8a depicts $\Delta\Phi$ for a significant wave height $H_S$ of 1 m. These wave heights correspond

to a wavelength $\lambda$ of approximately $25$ m, resulting in a typical deep-water wave steepness of $0.04$ (Heineke and Verhagen, 2009). Such conditions are representative of the prevailing small-wave regime during the measurement period (Deutscher Wetterdienst, 2023). Under these conditions, the wave-induced effect for RHI 1 on $\Delta\Phi$ is below $0.03°$ and for RHI 2 it is below $0.01°$ and and is therefore only a small part of the uncertainty of $\Phi$ determination. For most elevation intervals, $\Delta\Phi$ remains below $0.02°$, but exceptions occur in narrow intervals, for example between $-3.5°$ and $-2.5°$, where deviations exceed $0.1°$. In

contrast, for larger elevation intervals, wave-induced errors are typically approx. $0.01 - 0.03°$. At very shallow elevation angles (e.g., between $-1.5°$ and $-0.3°$), deviations again increase to around $0.03°$, with a further rise for smaller intervals at flatter angles. The RMSE for the conditions described is analysed to quantify the accuracy of the model in representing the wave-influenced data according to Eq. (13), which is shown in Fig. 8b. In intervals with steep angles, the RMSE amounts to $0.05°$, whereas for shallow angles it falls below $0.01°$. This indicates that the model assumptions are less robust to wave-induced

perturbations at steeper angles (i.e. shorter distances).


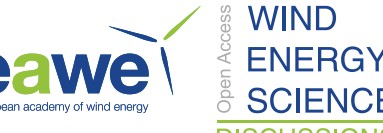


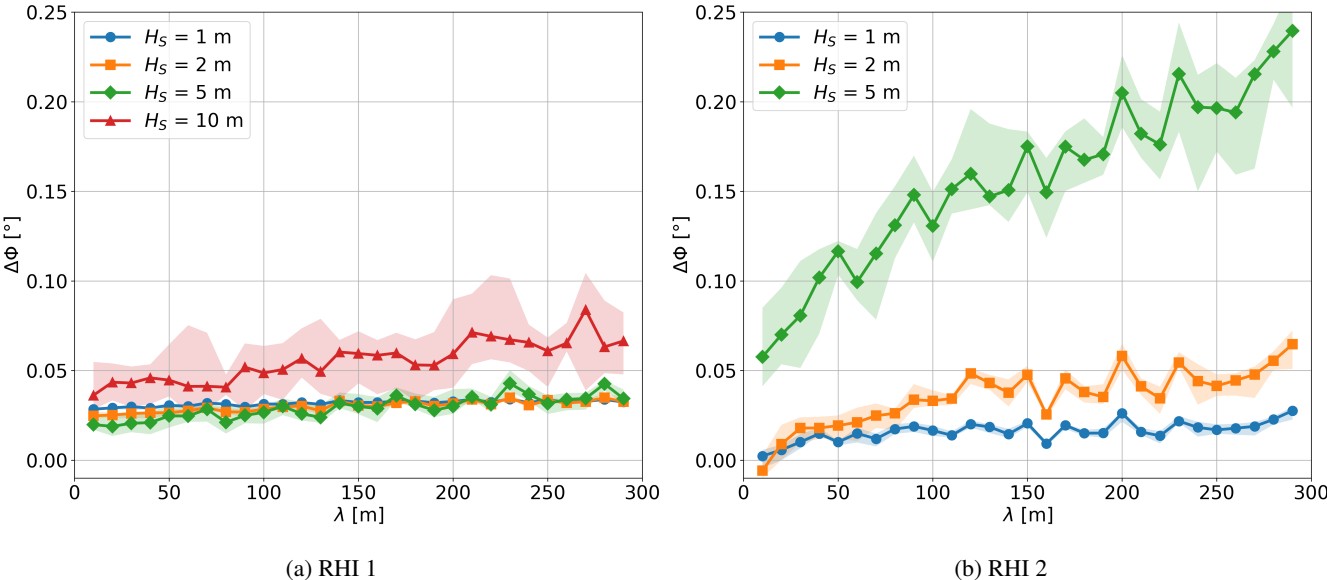

(a) RHI 1        (b) RHI 2

**Figure 9.** Illustration of $\Delta\Phi$ as a function different $\lambda$, based on a set of 10 random Rayleigh distributions for different $H_S$. The solid line indicates the median of the 10 distributions, and the shaded area represents the 10th–90th percentile range. (a) are results for the RHI 1 setting, (b) for the RHI 2 setting. In (b), results for $H_S = 10$ m are not shown, as the scale has been kept comparable to (a) to facilitate direct comparison.

Although the measurement campaign was conducted under $H_s$ of around 1 m, it is still important to assess the potential impact of larger waves on the measurements. To this end, further simulations were performed for various $H_\mathbf{s}$ and $\lambda$ under both RHI 1 and RHI 2 elevation interval configurations (see Fig. 9). However, the elevation and azimuth step sizes were reduced to limit the simulation duration. $H_\mathbf{s}$ of 1 m, 2 m, 5 m and 10 m are considered, with $\lambda$ between 10 m and 300 m. While some combinations of wave height and length are physically unrealistic, including them enables a theoretical exploration of how robust and sensitive the measurement method is under extreme or boundary-case conditions. This approach helps to identify potential limitations and guide improvements by revealing how the measurement system would respond beyond typical operational scenarios.

In the RHI 1 configuration, the influence of waves on $\Delta\Phi$ remains minimal at $H_\mathbf{s}$ of 1 m, 2 m, and 5 m, with deviations below $0.04°$. In contrast, the effect becomes pronounced at $H_\mathbf{s} = 10$ m, which corresponds to wave amplitudes typical of very stormy and extreme sea conditions, with maximum deviations reaching up to $0.1°$. It should be noted that SSL measurements are generally not performed under such conditions due to additional complications, such as high wind speeds and waves, and the possible operation of a wind turbine, which could induce platform tilting and other effects associated with high sea states. For small wave heights and short wavelengths, RHI 2 exhibits similar or even lower values than RHI 1. Notably, at RHI 2 with $H_\mathbf{s} = 2$ m and very short wavelengths, $\Delta\Phi$ occasionally becomes negative. For longer wavelengths at small wave heights of $H_\mathbf{s} = 2$ m, however, the influence of the waves can reach approximately $0.05°$ and should therefore be considered in the





uncertainty quantification for the RHI 2 setting. Furthermore, at $H_s = 5$ m and $\lambda = 125$ m (corresponding to a wave steepness of 0.04), $\Delta\Phi$ reaches $0.15°$. At $H_s = 10$ m in the RHI 2 configuration (not shown in the graph), $\Delta\Phi$ ranges between $0.4°$ and $0.6°$, suggesting that a reliable estimate of $\Phi$ under stormy conditions due to the resulting inaccurate determination of $r_w$ due to high wave amplitudes is associated with very high uncertainties.

### 3.2 Experimental results for the extended SSL method

This section presents the experimental evaluation of the extended SSL method using data from a scanning lidar WLS400S (Sect. 2.5), with data filtering as described in Section 2.6. The extended SSL method (model) is examined by comparing its predictions with measured data using the RMSE as an indicator of model agreement (Sect. 3.2.1). Subsequently, the SSL performance is evaluated for different RHI and PPI scan settings by repeating measurements over a period of time and comparing them with external data for the elevation offset (Sect. 3.2.2).

#### 3.2.1 RMSE of the extended SSL method (indicator of model quality)

As described in Sect. 2.2, the model parameters in Eq. (13) are determined based on SSL measurement data for $\varphi_{L'}$, $\theta_L$, and $r_w$ using optimisation methods. Specifically, the parameters are optimised using a non-linear adjustment calculation with the Levenberg–Marquardt algorithm. This process minimizes the sum of squared deviations between the model and observed elevation angles. The optimisation yields estimated values for the parameters: $\alpha$, $\beta$, $\Phi$, and $h_{\text{lidar}}$. The uncertainties of these parameters are quantified using the covariance matrix of the parameter estimation. The standard deviations, derived from the diagonal elements of this matrix, provide an indication of the statistical accuracy of the estimated values. However, due to the high number of values the analyses reveal very low uncertainty values for the individual parameters, so these uncertainties are considered negligible in subsequent analysis.

To assess the quality of the extended SSL model, the root mean square error (RMSE) between the measured data and the model predictions is evaluated. Figure 10a shows the raw measurement data for the RHI 1 measurement on 19.11.2023 at 22:22 UTC. The data, which represents a function of $\varphi_L$ and $r_w$, was filtered using the methods described in Sect. 2.6. There are no measured values from $45°$ to $180°$, as the laser beam is blocked by the wind turbine tower. In a first step, the model parameters, $\alpha$, $\beta$, $\Phi$ and $h_{\text{lidar}}$, are estimated by optimising the model to fit the measurement data using Eq. (13). The solution of this equation is interpreted as the result of a tilted system configuration caused by $\alpha$, $\beta$, and $\Phi$. By applying the inverse transformation, these effects are removed — effectively projecting the measurement back to a reference system with zero $\alpha$, $\beta$, and $\Phi$. Figure 10b shows the "corrected" measurement data. This backward transformation allows indirect validation of the estimated model parameters and makes it easier to identify measurement points that deviate from the expected behaviour (outliers).

The red line in Fig. 10b represents the idealised relationship between $\varphi_{L'}$ and $r_w$, as derived from the determined $h_{\text{lidar}}$. The deviation of the corrected data from this line is quantified by RMSE, which measures the average deviation between the modelled and observed values. This provides a benchmark for the model quality. In the example, the optimisation yields a pitch of $-0.11°$, a roll of $-0.07°$, an elevation offset of $-0.14°$, and a lidar height of $22.27$ m, with an RMSE of $0.03°$.





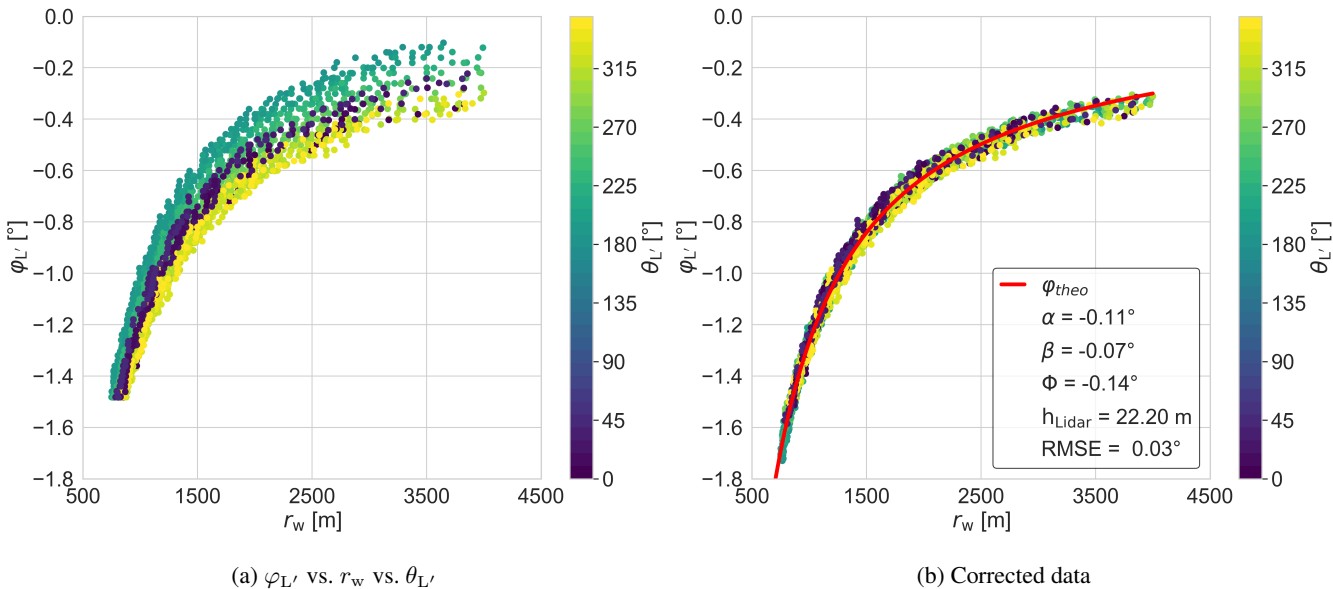

(a) $\varphi_{L'}$ vs. $r_w$ vs. $\theta_{L'}$                    (b) Corrected data

**Figure 10.** The dependence of $\varphi_{L'}$ on $r_w$ of an SSL measurement is shown using RHI 1 data. The colour scale represents different $\theta_{L'}$. (a) shows the measured $r_w$ compared to the internal lidar data. Based on these values, the Eq. (13) is determined by optimising the parameters for $\alpha$, $\beta$, $\Phi$ and $h_{\text{lidar}}$. (b) here, the $\varphi_{L'}$ data is corrected by taking into account the calculated tilt. The red line $\varphi_{\text{theo}}$ represents the model prediction based on the optimised parameters. The deviation between the measured values and the model curve is used to calculate the RMSE.

The dispersion of the values can thus be described well in the noise obtained by the uncertainties described in Sect. 3.1.1 and Sect. 3.1.2, as well as by minor variations in platform tilt recorded by the wind turbine's inclinometers over time (see Sect. 3.2.2).

### 3.2.2    Evaluation of the extended SSL method based on RHI and PPI scans

To analyse the SSL method, data collected over a period of ten hours is evaluated, as described in Section 2.5. Two distinct
RHI scan configurations are employed, along with a composite PPI scan routine consisting of six individual PPI scans at different elevation angles. For the PPI scans, the elevation angles in the lidar coordinate system $\varphi_{L'}$ range from $-3°$ to $-0.5°$ in increments of $0.5°$. The first RHI scan covers elevation angles from $-1.5°$ to $-0.3°$, while the second RHI scan includes angles from $-3°$ to $-1.5°$. For the analysis, only data within a range of up to $4000\,\text{m}$ are considered, as significant errors in sea surface distance estimation—based on the CNR-over-range approach occur beyond this limit and could not be adequately suppressed
by the filtering procedures described in Sect. 2.6. In addition, a retrospective analysis of the reconstructed measurement data, according to the methodology in Sect. 3.2.1 and illustrated in Fig. 10, allows the identification of values exhibiting strong deviations, which are subsequently removed as systematic outliers.



**Figure 11.** Evaluation of repetitive SSL scans using different patterns. Green RHI 1 ($\varphi_{\text{start}} : -1.5°$ to $\varphi_{\text{end}} : -0.3°$, step: $0.02°$), orange RHI 2 ($\varphi_{\text{start}} : -3°$ to $\varphi_{\text{end}} : -1.5°$, step: $0.02°$), and blue PPI ($\varphi_{\text{start}} : -3°$ to $\varphi_{\text{end}} : -0.5°$, step: $0.5°$). Crosses: uncorrected; points: $r_{\text{w}}$-corrected ($\epsilon_{\text{r}} = -37.5$ m, half pulse length). Shown: $\alpha$, $\beta$, $\Phi$, $h_{\text{lidar}}$, RMSE for 19–20 Nov 2023, under low wind and minimal platform motion. Right axes and dashed line display an external reference measurement; in panels 1–2: deviation of 10-min mean $\alpha$ and $\beta$ from wind turbine inclinometer; panel 3: $\Phi$ from external validation by DNV (2022).



SSL measurements vary across different scan configurations; however, applying a correction to the determination of $r_\mathrm{w}$ reduces these differences. Figure 11 presents the SSL evaluation results over time for pitch $\alpha$, roll $\beta$, elevation offset $\Phi$, lidar

height $h_\mathrm{lidar}$, and RMSE across all configurations: RHI 1 (green), RHI 2 (orange), and PPI (blue). Crosses indicate uncorrected values from individual SSL scans, while points represent measurements in which the distance — derived from the CNR-over-range data (see Sect. 2.3) — was corrected by $\epsilon_\mathrm{r} = -37.5$ m, corresponding to half the pulse length before the dipping point. The sequential repetition of measurements under consistent environmental conditions serves to validate the results and enhances their significance. Depending on the configuration, each scan lasts approximately 25–30 minutes.

Regardless of the SSL scan settings, pitch and roll exhibit minimal fluctuations throughout the measurement period. The top two panels in Fig. 11 depict the temporal evolution of pitch and roll. As the scanning lidar is mounted on the wind turbine platform, mechanical fluctuations occur, particularly during operation of the turbine. Therefore, to enable meaningful comparisons between measurements, it is essential to account for possible platform movements at the time of acquisition. The deviations of the wind turbine's inclinometer readings from their respective 10-min mean values are shown on the right-hand

axis to provide a measure of platform inclination. The pitch and roll values determined in our measurements cannot be directly compared with the absolute values of the wind turbine's inclinometer, but the shown deviations from the mean indicate that only slight platform movements were recorded by the inclinometer throughout the entire measurement period. The measurement was scheduled in such a manner that the wind turbines were not operational due to wind speeds of less than $4$ m/s. The 10-min average inclinometer deviations in pitch and roll remained below $0.020°$ and $0.025°$, respectively.

Likewise, pitch and roll values derived from the SSL method (left axes) exhibit only small variations. The corresponding mean values and standard deviations over time are listed in Table 1. The differences between configurations are negligible: the mean pitch values are $-0.116°$ for and RHI 2, and $-0.123°$ for PPI. Applying the distance correction had no significant influence on the pitch or roll values. A similar pattern can be observed for roll, with mean values of $-0.085°$ for RHI 1, $-0.084°$ for RHI 2, and $-0.068°$ for PPI. Overall, the differences between the various settings remain below $0.02°$, which is

well within the maximum alignment accuracy of $0.02°$ due to the mechanical gears in the scanning lidar.

The third panel in Fig. 11 presents the elevation offset, a constant parameter of the measuring device that should remain unchanged over time, even in the presence of platform-induced tilts. The time averages of $\Phi$ differ notably between configurations: $-0.140°$ for RHI 1, $-0.282°$ for RHI 2, and $-0.148°$ for PPI, with a maximum difference of $0.127°$ (see also Table 1). The standard deviations over time for each setting are also larger, compared to pitch and roll standard deviation. This variability

could be explained by the fact that, as evident from the preceding analyses, $\Phi$ is particularly sensitive to errors, such as those arising from inaccurate $r_\mathrm{w}$ measurements (see Sect. 3.1.1 and Sect. 3.1.2).

If the CNR drop is assumed to occur not at the inflection point of the fitted sigmoid describing the CNR-over-range curve at the water surface, but half a pulse length earlier ("corrected" data), the difference between RHI 1 and RHI 2 is significantly reduced. Applying a $\epsilon_\mathrm{r}$ of $-37.5$ m (distance correction) brings the elevation offset of the two RHI configurations closer together,

as indicated by the green and orange points in Fig. 11. The PPI data also shift into a similar range, yielding $-0.087°$, although the remaining maximum difference to RHI 2 is still $0.037°$. For validation, $\Phi$ obtained from an independent measurement by an external consultancy at the same location with the same lidar device several months earlier is considered (DNV, 2022). Here,



**Table 1.** Mean values and standard deviation of the SSL parameters for RHI 1, RHI 2 and PPI ($\alpha$, $\beta$, $\Phi$, $h_{\mathrm{lidar}}$ and RMSE) over the measurement period from 19 November 22:00 UTC to 20 November 8:00 UTC 2023. Because $h_{\mathrm{lidar}}$ cannot be assumed to be a constant value, due to the tide, the minimum, maximum and the maximum difference in the period are shown here. The SSL results are also shown with a $r_{\mathrm{w}}$ correction of $\epsilon_{\mathrm{r}} = -37.5$ m.

| Variable | Statistic | RHI 1 | RHI 1 + $\epsilon_{\mathrm{r}}$ | RHI 2 | RHI 2 + $\epsilon_{\mathrm{r}}$ | PPI | PPI + $\epsilon_{\mathrm{r}}$ |
|---|---|---|---|---|---|---|---|
| $\alpha$ | Mean | $-0.116°$ | $-0.117°$ | $-0.123°$ | $-0.123°$ | $-0.123°$ | $-0.127°$ |
| | Std. Dev. | $0.005°$ | $0.005°$ | $0.007°$ | $0.007°$ | $0.006°$ | $0.005°$ |
| $\beta$ | Mean | $-0.085°$ | $-0.086°$ | $-0.082°$ | $-0.081°$ | $-0.068°$ | $-0.068°$ |
| | Std. Dev. | $0.017°$ | $0.017°$ | $0.006°$ | $0.006°$ | $0.006°$ | $0.007°$ |
| $\Phi$ | Mean | $-0.140°$ | $-0.116°$ | $-0.282°$ | $-0.124°$ | $-0.148°$ | $-0.087°$ |
| | Std. Dev. | $0.014°$ | $0.013°$ | $0.033°$ | $0.029°$ | $0.018°$ | $0.022°$ |
| $h_{\mathrm{lidar}}$ | Minimum | $21.655°$ | $20.402°$ | $23.779°$ | $20.705°$ | $22.768°$ | $20.555°$ |
| | Maximum | $22.941°$ | $21.618°$ | $24.684°$ | $21.510°$ | $23.992°$ | $21.795°$ |
| | Range | $1.285°$ | $1.217°$ | $0.904°$ | $0.805°$ | $1.225°$ | $1.239°$ |
| RMSE | Mean | $0.041°$ | $0.041°$ | $0.051°$ | $0.052°$ | $0.058°$ | $0.057°$ |
| | Std. Dev. | $0.013°$ | $0.013°$ | $0.002°$ | $0.002°$ | $0.014°$ | $0.019°$ |

the given $\Phi$ of $-0.112°$, determined via hard target CNR mapping, is shown on the right y-axis of third panel in Fig. 11 and is notably closer to the distance-corrected results. Overall, this value confirms that the elevation offset can be reliably determined
using the SSL method.

The height of the lidar above the sea level $h_{\mathrm{lidar}}$, shown in the fourth panel in Fig. 11, is not a parameter directly required for aligning the scanning lidar. In our measurements, the exact height of the lidar above the sea surface was unknown and was therefore included as an optimisable variable in Eq. (13). Because the height of the lidar above the surface can vary over time due to local conditions, such as tidal fluctuations, it should not be assumed constant. Over the ten-hour measurement
period, a temporal variation in $h_{\mathrm{lidar}}$ was observed. The $h_{\mathrm{lidar}}$ values of RHI 1, RHI 2 and PPI initially differ significantly, but they gradually converge after applying the distance correction, which in turn provides additional evidence that the correction procedure contributes to improving the estimation of $\Phi$.

The RMSE displayed in the bottom panel quantifies the deviation between the optimised model and the measured data (see Sect. 3.2.1), serving as an indicator of the model's uncertainty. Mean RMSE values of $0.041°$ for RHI 1, $0.054°$ for RHI 2 and
$0.058°$ for PPI demonstrate that the extended SSL model generally provides a good fit to the measurement data.





# 4 Discussion

The aim of this investigation was to introduce and extend the SSL method in order to examine whether it could be used to reliably estimate the value of the elevation offset $\Phi$. In addition, the paper assesses the factors that influence uncertainties in SSL-derived alignment parameters.


- *Stability and accuracy of pitch and roll determination* The experimental results show that the determination of pitch and roll angles using the extended SSL method is very reliable under the conditions prevailing in this study (Table 1 and Fig. 11). In calm wind conditions, with wind speeds slower than $4 \text{ m/s}$ and significant wave heights of around 1 m, reproducible results were achieved regardless of whether RHI or PPI approaches were used. The small RMSE values confirm that the model accurately describes the measured data. Even with a line-of-sight obstruction of $40°$ to $180°$,

where no measurements could be taken, the results remained consistent. The influence of distance errors or waves on the determination of these two parameters also proved to be negligible, as described in Sect. 3.1.1 and shown in Fig. 7a.

- *Sensitivity and influencing factors on the elevation offset* In contrast to pitch and roll, the elevation offset is significantly more sensitive to measurement settings and environmental conditions. However, unlike in previous studies (Rott et al., 2022; Gramitzky et al., 2024), $\Phi$ could be derived directly from the extended SSL method introduced in this paper. The

most important influencing factors are the scanning strategy – in particular the interval of the elevation angles – which are particularly evident after applying a distance correction (see Table 1 and third panel of Fig. 11). The accuracy of $\Phi$ is therefore directly related to the exact determination of the distance to the sea surface $r_{\text{w}}$. However, if we compare the corrected $\Phi$ results to the validation measurement by DNV (2022), our results are within the same range of values. This performance confirms that the elevation offset can be determined using the extended SSL method.

- *Determining the distance to the water surface* The greatest uncertainty in determining $\Phi$ and height of the lidar above sea surface $h_{\text{lidar}}$ lies in accurately detecting the point at which the laser beam enters the water surface $r_{\text{w}}$. Factors likely to influence this uncertainty are the pulse length and pulse shape of the laser and the logarithmic scaling of the CNR values. The SSL results indicate that the inflection point of the sigmoid function in the CNR over range curve is very unlikely to correspond exactly to the actual point of entry of the laser beam into the water. Accurate modelling of

$r_{\text{w}}$ requires detailed knowledge of the weighting function (see Eq. (15) in Sect. 2.3), which depends on pulse shape, signal processing, and device type, and is therefore difficult to determine. Due to the very low signal strength near the water surface, the modelling is particularly sensitive to assumptions about the tail of the weighting function. For this study, a simplified constant distance correction based on half the probe volume length was applied, representing a rough but practical approximation. Future research could address this problem by performing experiments using targets with

known distances. Moreover, repeating the SSL measurement for different pulse length would help to systematically quantify their influence on the the determination of $r_{\text{w}}$. However, these were not available to the authors by the time of writing this publication. While this distance correction remains negligible for pitch and roll, it has a significant effect on $\Phi$ and $h_{\text{lidar}}$. However, since the inflection point of the sigmoid function is a very robust method for determining a fixed



point in the falling CNR curve, it can still be a useful approach to determine the distance to the water surface in this way and correct the distance accordingly. An initial correction assumption — setting the entry point at approximately half the pulse length before the inflection point — led to a significant improvement in the agreement with the external reference measurement of $\Phi$ and reduced the differences between the various settings (see Table 1 and third panel of Fig. 11). In addition, the $h_{\text{lidar}}$ is not constant over time due to the influence of tides. In this measurement campaign, $h_{\text{lidar}}$ was assumed to be constant for periods of about $30$ min, but was determined individually for each SSL evaluation, yielding fluctuations in the range of $0.805$ m to $1.285$ m (see Table 1 and forth panel in Fig. 11).

– *Wave influence* The measurement campaign took place at small wave amplitudes ($< 1$ m significant height) and wind speeds below $4$ m/s (Sect. 3.2.2), so the influence of waves is under $0.01°$ for RHI 2 settings and under $0.03°$ under these conditions, as confirmed by the modelling results (Fig. 8). However, at steeper elevation angles and very short interval ranges, larger uncertainties in the elevation offset occur. The effect of waves on pitch and roll determination can be neglected, as indirectly indicated by the distance error evaluations (Fig. 7a). For larger wave heights, a notable influence occurs only at the RHI 1 setting (elevation angles between $-1.5°$ to $-0.3°$) at $10$ m (waves typically observed in very stormy conditions), whereas at the RHI 2 ($-3°$ to $-1.5°$) setting, wave heights from as low as $2$ m can already significantly increase the uncertainty of the elevation offset, depending on the wavelength. At a wave height of $5$ m, the modelling suggests that determining the elevation offset, typically on the order of tenths of a degree, becomes challenging due to deviations caused by the waves. A simplified wave model was used for the analysis, assuming one-dimensional, sinusoidal waves propagating in the x-direction with Rayleigh-distributed amplitudes and constant frequency, without cross waves or spectral superposition. This idealized approach does not aim to fully represent the complex behaviour of real waves, but focuses on the effects of different amplitudes and wavelengths. The analysis shows that higher waves have the largest influence on $\Delta\Phi$, with RHI 2 settings being more affected than RHI 1. These simplifications allow an initial estimate of the impact of water waves on the measurement results. Random superposition of real waves may introduce additional noise, which can be reduced statistically with sufficient measurements and does not necessarily lead to systematic distortion of the results.

– *Practical recommendations* The results show that flatter elevation angles generally enable more robust measurements with lower uncertainty. The RHI 1 with elevation angles between $-1.5°$ to $-0.3°$ setting was particularly suitable in this study, which allowed distance measurement up to $4000$ m a lidar height of $20$ m. At these settings, uncertainties in the elevation offset in the range of $0.03°$ to $0.04°$, due to distance errors of half the pulse length and wave uncertainties (see Fig. 6a and Fig. 8a). Further flattening of the angles is not recommended, as tilting of the laser beam would prevent it from hitting the water surface in all directions. In our measurement data the measurement distance limit was approximately $4000$ m, as beyond this distance the CNR signal decrease prevents reliable determination of the entry point. In general, SSL measurements should be avoided at high wind speeds when the lidar is mounted on the transition piece of a wind turbine and the turbine is in operation, as changing thrust and waves are likely induce changes in turbine tilt during the scans used in a a SSL. In addition, larger wave amplitudes also influence the results (Sect. 3.1.2).





- *Outlook and transferability* The obtained results can be transferred to other scanning lidar systems, provided the laser beam entry point is known with low uncertainty. The SSL method relies on simplifications valid only for small pitch, roll, and elevation angles. However, the method can also be transferred to larger pitch, roll, and elevation angles, if the complete Eq. (12) is used instead of the simplified Eq. (13), which also enables its application to drones (see Sect. 3.2.1). However, the non-simplified version requires more complex optimisation, which did not yield any discernible advantage in this measurement campaign due to the small elevation angles of less than $\|3°\|$. Further development to determine the exact point of entry of the laser beam into the sea surface would significantly reduce the uncertainties in the elevation offset and lidar height. Furthermore, the present uncertainty analysis considers different settings, but does not take into account that no discrete LOS scans were performed during the RHI and PPI scans, but rather that angle ranges were scanned continuously, which could explain differences between the RHI and PPI results in the elevation offset and the lidar height and also some of the scatter in the measurement result (RMSE). This aspect should be investigated in more detail in future work.

## 5   Conclusions

The aim of this study was to extend the Sea Surface Levelling (SSL) method, which had previously been limited to pitch and roll estimation, in order to also determine the static elevation offset and to establish a framework for uncertainty analysis. To this end, we applied the extended SSL approach to 10 hours of offshore measurements with a scanning lidar (Vaisala WindCube WLS400S) mounted on the transition piece of a wind turbine in the German North Sea. The method incorporated both RHI and composite PPI scan configurations and explicitly accounted for small pitch, roll, and elevation angles as well as Earth's curvature. With this setup, the extended SSL method successfully yielded consistent and reproducible values for pitch, roll, and elevation offset.

   The determination of pitch and roll values proved to be stable and accurate throughout the campaign. Although the elevation offset was more sensitive to measurement and environmental factors, applying a distance correction resulted in values that were consistent with the validation data provided by external experts (DNV, 2022), clearly confirming the validity of the method. The main source of uncertainty originates from determining the exact point of entry of the laser into the water $r_\mathrm{w}$, which can be partly mitigated by applying a half-pulse-length correction ($\epsilon_\mathrm{r} = -37.5$ m). Uncertainty analysis shows that using flatter elevation angles between $-1.5°$ and $-0.3°$ (RHI 1 setting) is well suited to achieving low-uncertainty results when the lidar is positioned approx. 20 m above the water surface. In this configuration, $r_\mathrm{w}$ corresponds to ranges of approximately 795–3820 m for $\alpha = \beta = \Phi = 0$. Although these distances vary when the tilt angles are taken into account, the influence of the uncertainties in $r_\mathrm{w}$ becomes less critical at these distances. However, under the environmental conditions of the campaign, $r_\mathrm{w}$ could not be reliably detected beyond 4000 m with the device. Wave effects were minor during the test campaign but may become significant at steeper angles or under rougher sea states. Wave-induced uncertainties of approximately $0.03°$ for RHI 1 and $0.02°$ for RHI 2 can explain part of the noise observed in the RMSE values. Other relevant factors are uncertainties in $r_\mathrm{w}$ and





platform inclination. To minimise platform tilt and high-frequency vibration effects, measurements should not be carried out with the lidar mounted on a transition piece during turbine operation.

Overall, the extended SSL method can be transferred to other scanning lidar systems and, without simplifying assumptions, can also be expanded to include larger elevation angles. Future research should prioritize refining the determination of the laser entry point. Although correcting the distance to the sea surface by half the pulse length proved to be a successful and reasonable

initial approach to mitigate discrepancies observed across different settings, a more precise measurement of the distance to the water surface could significantly reduce uncertainties.

*Code and data availability.* The extended SSL analysis code and an example dataset are being prepared for public release.

*Author contributions.* The study was primarily conducted by KG, who was responsible for the conceptual design, theoretical considerations, development and execution of the measurement campaign, data analysis, and preparation of the manuscript and figures. LP provided main

supervision and overall guidance throughout the project. FJ and LP contributed to the theoretical and conceptual development of the methods and supported the refinement of the methodology; FJ also held a supervisory role in analysing and discussing the results. TH contributed through discussions on the measurement campaign and supported its execution. DC and JKL provided substantial input through valuable comments, extensive discussions, and critical advice, which significantly strengthened the work. All authors contributed through discussions, validation of the results, and considerable effort in reviewing the manuscript feedback on the manuscript.

*Competing interests.* One of the co-authors is a member of the editorial board of Wind Energy Science.

*Acknowledgements.* This work was supported by the German Federal Ministry for Economic Affairs and Climate Action (Window project, funding code: 0324159). KG gratefully acknowledges support from the Heinrich Böll Foundation. Special thanks go to ENBW (Energie Baden-Württemberg AG), in particular to Jens Riechert for his support during the measurement campaign and for valuable technical discussions. We also thank our colleagues at Fraunhofer IEE, Janis Musche and Luis Michaelis, for their assistance with the measurement campaign

and IT support. The authors used ChatGPT and DeepLwrite to improve grammar and style; all results and text were originally produced by the authors without using generative AI and the authors also reviewed and edited any suggestions for improvements made by ChatGPT.



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
