# Peer review of "Alignment of Scanning Lidars in Offshore Campaigns - an Extension of the Sea Surface Levelling Method"

_Wind Energy Science, 2025_

## Referee Comment (RC1)

Alignment of Scanning Lidars in Offshore Campaigns –
an Extension of the Sea Surface  Levelling Method

https://doi.org/10.5194/wes-2025-191

Review conducted by Elliot Simon (DTU), with support from Mathieu Pellé (DTU)

November 6, 2025

**General comments:**

This paper is well structured and well written. The research outcomes make a significant contribution towards enabling and verifying the correct alignment of scanning lidars in offshore installations. I am well-acquainted with the preceding PPI method (Rott et al. 2022) and the conference paper introducing the RHI approach (Gramitzky et al. 2024). Having already applied these methods in my own field campaigns, I expect the RHI-SSL technique to become the new standard procedure in offshore and nearshore scanning lidar installations where hard targets are unavailable. Although drone-based hard-targeting may still be advantageous in certain cases. While the paper is quite lengthy and could be condensed, I appreciate the authors' efforts to include sensitivity analyses and uncertainty estimations, as well as their reflections on the assumptions and limitations of the simplified models used. The work as presented here is thorough, scientifically rigorous, and fits the criteria for publication. My recommendation is that the paper be accepted following the authors' consideration of the following comments and implementing any necessary revisions.

**Specific comments:**

Line 15:  "$0.03\,^{\circ}$-$0.04\,^{\circ}$"

Change to $0.03\,^{\circ}$ to $0.04\,^{\circ}$ for consistency.

Line 49:  "precise alignment of the laser"

Better to define as the positioning of the laser beam or laser light emission.

Line 55:  "A target accuracy of about $0.0255\,^{\circ}$ has been shown to be achievable"

The achievable pointing accuracy will also depend on lidar manufacturer and mechanical/optical tolerances within the specific lidar unit.

Lines 60-61:  "drone positional uncertainty which can be significant at longer ranges"

This may be true if scanning the drone with CNR mapping as Oldroyd et al. 2024 demonstrated, but when following the method in Thorsen et al. 2023, where the lidar is in staring (LOS) mode and the drone if flown into the beam path, the further away the drone is flown, the smaller the positioning uncertainty becomes.

Lines 61-62:   "In recent tests, Oldroyd et al. (2024) reported uncertainties of up to 0.17° under high wind conditions, though values closer to 0.05° may be achievable in calmer conditions"

The following work has demonstrated significantly lower uncertainties in elevation, pitch and roll estimations in both the onshore and offshore campaign, similar to the values suggested as achievable.
Thorsen, G. R., Simon, E., & Clausen, E. H. (2023). *Drone-based scanning lidar pointing calibration (D4.4)*. DTU Wind and Energy Systems.

Lines 95-100: "The structure of the paper is as follows.."

This may be a personal opinion, but I find this paragraph unnecessary in a journal paper.

Line 111:   "Possible causes include.."

Also due to following errors in the scan head's positioning due to e.g. issues with the motor encoder, hall-effect sensor, mechanical misalignment, or motion control software or software parameters.

Line 136:   "in the context of scanning lidar devices, pitch and roll are usually small enough"

Could you provide a range of values where this assumption is valid?

Line 166:   "SSL is a method of determining the beam alignment calibration of a scanning lidar"

I wouldn't necessarily refer to this as a calibration method, as there is no comparison against an agreed standard reference, e.g. a calibrated theodolite also measuring the distance to the water surface. I would propose calling it a pointing verification method instead.

Line 199:   "neglecting $h_{earth}$"

I don't see this defined earlier. Could this be $z_{earth}$?

Lines 216-18: "The difference is that in RHI scanning, the beam is continuously moved through elevation angles, and the returned signal is an integration between two elevation steps. Therefore a small angular resolution should be chosen. In contrast, PPI scanning is performed at discrete elevation angles in step-stare mode."

Both methods (RHI and PPI) can be performed in continuous scan or step-stare mode. Is this reasoning due to a requirement for multiple elevation angles to be measured? i.e. multiple PPIs at different elevation angles vs. a single RHI at a single azimuth angle?

Line 228-29: "The inflection point of this function is then considered to be the point at which the laser beam enters the water surface"

Has this been verified experimentally? It would be interesting to carry out a small experiment using a calibrated reference e.g. theodolite/total station to verify the actual distance to the water surface, and compare against the estimate obtained via processing the lidar's CNR values. DTU would be interested in partnering on this exercise.

Line 247: "approx." would be better written out as approximately. This also appears on Lines 420, and 629.

Line 249: "middle of probe volume" should be "middle of the probe volume"

Line 284: "his model" should be "This model"

Line 295: "s the water depth"

Is the assumption in the wave model that this is deep water? I cannot find a mention of the water depth used. It appears that the site used in this study has a water depth of 35m.

Line 316-17: "This minimizes the effects of platform inclination caused by the thrust of the wind turbine during the measurement"

We have observed persisting platform motion long (i.e. hours) after the turbine is stopped. Mainly vibrations in higher energy cycles with a zero-mean displacement, although this depends on foundation type, stiffness and sea state. These get averaged out on longer timescales but fundamentally do influence the instantaneous lidar beam positioning. It would be nice to provide guidance on how long the turbine should be stopped for, and an indication of the higher frequency motion within the 10-minute inclinometer data (e.g. standard deviation).

Line 322-323: "(RHI 1) ranged from start elevation angle $\phi$ start $-1.5\circ$ to end elevation angle $\phi$ end $-0\circ$"

This may be a typo, RHI 1 appears to end at -0.3 degrees (e.g. Line 378)

Line 323-24: "The azimuth resolution for both scans was $5\circ$."

Please define the number of RHI scans in one sequence. This was slightly confusing as I wouldn't expect an RHI scan to have an azimuth resolution.

Line 328: "pulse length of 75 m"

This appears throughout the paper. I assume you mean probe length or range resolution as Vaisala call it. Pulse length is the duration of the emission of the laser pulse (normally in nanoseconds). A probe length of 75m represents a pulse length of 500ns.

Additionally, you may want to state that this is the smallest probe volume (pulse length) option on the 400S. Since other lidar systems (e.g. 100/200S or other manufacturers) support shorter pulse lengths, which may be preferable in the SSL approach.

Line 335: "Data filtering"

It would be helpful to indicate how much of the data is being removed in each step of the filtering process.

Line 346: "This range should be adjusted depending on meteorological conditions"

What conditions do you expect to influence the performance of this filter? Aerosol concentration?

Line 394:      Figure 7a

If possible, change the way this information is shown because the markers are stacked atop each other and difficult to interpret.

Line 436-437: "It should be noted that SSL measurements are generally not performed under such conditions"

It may not be straightforward for the lidar operator to identify periods with low wave and turbine motion to run the SSL scans. This could be done manually or in an automated manner using forecasts or live measurements if available, however I have often seen SSL scans programmed to execute on a routine schedule regardless of conditions present at the site. There are also sites where these conditions are rarely met, even during installation and decommissioning.

Line 488:      Figure 11, "Right axes and dashed line display an external reference measurement"

I would be very interested to see the sub 10-minute variability (e.g. error bars of standard deviation) of the lidar/platform motion. I assume this data is taken from a period where the turbine is shut down for a longer period but this may not be the case.

Line 522:      "several months earlier"

I agree that the elevation offset should not change with time, although I have observed this happening due to mechanical or software faults. Repeating this as a post-campaign check would strengthen the belief that it has not changed.

Line 595:      "up to 4000 m a lidar height of 20 m."

"At a lidar height of 20 m?"

Line 642:      "The extended SSL analysis code and an example dataset are being prepared for public release"

It will be greatly appreciated to share the validated processing code with the community and include it together with this publication.